# Humidity effects on the detection of soluble and insoluble nanoparticles in butanol operated condensation particle counters

Christian Tauber[1], Sophia Brilke[1], Peter Josef Wlasits[1], Paulus Salomon Bauer[1], Gerald Köberl[1], Gerhard Steiner[1,2,3], and Paul Martin Winkler[1]

[1]Faculty of Physics, University of Vienna, Boltzmanngasse 5, 1090 Vienna, Austria
[2]Institute for Ion Physics and Applied Physics, University of Innsbruck, Technikerstraße 25/3, 6020 Innsbruck, Austria
[3]Grimm Aerosol Technik Ainring GmbH & Co Kg, Dorfstraße 9 Ainring, 83404 Ainring, Germany

**Correspondence:** Christian Tauber (christian.tauber@univie.ac.at)

**Abstract.** In this study the impact of humidity on heterogeneous nucleation of n-butanol onto hygroscopic and nonabsorbent charged and neutral particles was investigated using a fast expansion chamber and commercial continuous flow type condensation particle counters (CPCs). More specifically, we measured the activation probability of sodium chloride (NaCl) and silver (Ag) nano-particles by using n-butanol as condensing liquid with the size analyzing nuclei counter (SANC). In addition, the cut-off diameters of regular butanol based CPCs for both seed materials under different charging states were measured and compared to SANC results. Our findings reveal a strong humidity dependence of NaCl particles in the sub-10 nm size range since the activation of sodium chloride seeds is enhanced with increasing relative humidity. In addition, negatively charged NaCl particles with a diameter below 3.5 nm reveal a charge enhanced activation. For Ag seeds this humidity and charge dependence was not observed, underlining the importance of molecular interactions between seed and vapor molecules. Consequently, the cut-off diameter of a butanol based CPC can be reduced significantly by increasing the relative humidity. This finding suggests that cut-off diameters of butanol CPCs under ambient conditions are likely smaller than corresponding cut-off diameters measured under clean (dry) laboratory conditions. At the same time, we caution that the humidity dependence may lead to wrong interpretations if the aerosol composition is not known.

## 1 Introduction

In the atmosphere nano-particle formation by gas-to-particle conversion has been observed in a variety of locations and conditions (Kulmala et al., 2004). On a global scale, it is seen as an important source controlling the number size distribution of atmospheric aerosols (Kulmala et al., 2014). Thereby, a phase transition from the gaseous to the liquid or solid state occurs in the presence of supersaturated vapors. This process of nucleation arises homogeneously from gas molecules only, or heterogeneously by the formation of vapor clusters on preexisting particles (Strey et al., 1986; Winkler et al., 2008b; Kangasluoma et al., 2016; Ferreiro et al., 2016; Tauber et al., 2018). These nano-particles then can grow up to a size at which they act as cloud condensation nuclei (CCN) (Spracklen et al., 2008; Merikanto et al., 2009). Thereby nanometer-sized particles contribute to the indirect radiative forcing, thus influencing the Earth's climate (IPCC, 2013).

Heterogeneous nucleation in the atmosphere takes place for seed particles and vapor molecules of different chemical composition. Aerosol particles acting as seeds in the heterogeneous nucleation process have different physicochemical surface properties such as charging state, wettability, shape or size. Interactions between the seed and the vapor molecules as well as solubility play an important role (Kupc et al., 2013; McGraw et al., 2012, 2017). Atmospheric aerosol can contain hygroscopic salts and is therefore sensitive to relative humidity which can contribute to a phase transition within the aerosol particle.

A highly hygroscopic and well characterized example is sodium chloride (NaCl) which mainly originates from sea spray (Biskos et al., 2006; Krämer et al., 2000; Clarke et al., 2003; Lawler et al., 2014; Zieger et al., 2017). Water vapor strongly influences the phase of the hygroscopic NaCl particles which under dry conditions exhibit a solid crystalline structure. At high relative humidity (RH), salt particles take up water and form saline droplets with increased volume which affects their physical, chemical and optical properties. In contrast to increasing RH, aqueous saline aerosol particles shrink with decreasing relative humidity. Thereby the water evaporates and the seed crystalizes (Martin, 2000; Biskos et al., 2006).

In a recently published paper by Tauber et al. (2018), it has been shown that the presence of monoatomic ions significantly lowers the energy barrier for the heterogeneous nucleation of n-butanol compared to neutral seeds. It is known from previous studies that ion induced nucleation is highly affected by the chemical structure of the condensing vapor and seed ion properties like charge state (Iida et al., 2009; Kangasluoma et al., 2016). This process comes into play especially in the sub-3 nm size range at which condensation-based nanoparticle detection methods have their lower detection limit. Thereby the initial charge state of aerosol particles with diameters within this size range can influence the detection efficiency.

In the atmosphere usually more than a single species of vapor molecules contributes to the vapor-liquid nucleation. Both processes, gas-to-particle conversion and the existence of insoluble particles in the atmosphere, can initiate heterogeneous nucleation processes (Kulmala et al., 2004; McMurry et al., 2005; Kuang et al., 2010, 2012; Wang et al., 2013). A commonly used technique to measure preexisting particles in the atmosphere is condensation particle counting, which often use n-butanol as working fluid (McMurry, 2000). However, under ambient conditions significant amounts of water vapor usually enter the condensation particle counter (CPC) and can modify its detection behavior. Previous studies on binary heterogeneous nucleation of n-propanol-water vapor mixtures on sodium chloride (NaCl) particles indicated a decrease of the activation barrier in accordance with the theory of binary heterogeneous nucleation and reported a soluble-insoluble transition in the particle activation behavior (Petersen et al., 2001). Butanol is chemically similar to propanol and commonly used as working fluid in CPCs. Therefore the process of heterogeneous nucleation of n-butanol and water vapor on nanometer sized particles is of high relevance in ambient nanoparticle characterization studies. In addition, pulse height analysis conducted by Hanson et al. (2002) for sulfuric acid particles revealed a strong dependence of the particle counter's response on chemical composition and water vapor. The size resolved chemical composition measurements of nanoparticles from reactions of sulfuric acid with ammonia and dimethylamine, investigated by Chen et al. (2018), suggest that small, acidic and newly formed particles can affect the physicochemical properties during the cluster formation process and thereby enhance early particle growth. This indicates a chemical cluster composition and working fluid dependent activation behavior at the lower detection limit of the used CPC.

Over the last years various studies investigated the onset saturation ratio and the nucleation temperature for different seeds and vapors. These studies include measurements by Chen and Tao (2000) for water on SiO2 and TiO2, and Schobesberger

et al. (2010) for n-propanol on silver (Ag) and sodium chloride seeds. All studies show that the expected temperature trend agrees with theory, except for the nucleation of n-propanol on NaCl particles. In this specific case an opposite temperature trend of the onset saturation ratio compared to the Kelvin equation in the temperature range from 262 K to 287 K was recorded (Schobesberger et al., 2010). Despite the common use of butanol as working fluid for the detection of ambient nanoparticles, comparatively little research on the fundamental aspects of butanol nucleation has been done in the sub-10 nm size range, e.g. Barmpounis et al. (2018) and Sem (2002). For DEG based CPCs Iida et al. (2009) and Kangasluoma et al. (2013) investigated the temperature and humidity dependence of the working fluid on the particle activation. Here, we investigate heterogeneous nucleation of n-butanol vapor on differently sized seeds in the sub-10 nm diameter range depending on relative humidity, charge state and nucleation temperature. We aim at getting a better understanding of the effect of RH and initial charging states on the nucleation process in CPCs with the focus on sub-5 nanometer particles. Thereby the relative humidity effect and charge dependence on the lower particle detection limit can be analyzed. The results help to improve the understanding of heterogeneous nucleation of n-butanol vapor on soluble and insoluble seeds. By that the acting mechanism during the nucleation process can be analyzed and used to explain changes of the counting efficiencies in commerical CPCs (Ankilov et al., 2002).

## 2 Experimental Section

We examined the nucleation probability and counting efficiency of n-butanol based CPCs as shown schematically in Figure 1 and Figure S1 in the supporting information. Ag and NaCl particles were generated in a tube furnace (Scheibel and Porstendörfer, 1983) which was supplied with synthetic air (ALPHAGAZ 1 AIR, $>= 99.999\%$ (5.0), Air Liquide) or particle-free dry compressed air. The flow carrying the polydisperse aerosol was kept constant at 3 L/min through the Americium 241 (Am-241) charger and the nano differential mobility analyzer (nDMA) for both experimental approaches. By applying positive or negative voltage to the nDMA a monodisperse negatively or positively charged particle fraction was selected. By placing an Am-241 charger either at the Size Analyzing Nuclei Counter (SANC) or the CPC inlet the size selected particles were neutralized. Subsequently, ions that form inside the charger were removed using a home-built ion precipitator (ion trap). By applying a voltage of $\pm 500$V to a tubing which is cut in two half pipes and separated by a non conductive material all charger ions were removed. An adjustable well-defined flow of humid air joined the aerosol flow after the nDMA in order to vary the humidity before the CPC inlet flow. Accordingly, the carrier gas was humidified by passing it through a diffusion type humidifier (see Figure 1). The relative humidity of the carrier gas was monitored and recorded throughout every measurement. The recorded RH values were stable within $\pm 1\%$ which is below the sensor accuracy. Two humidity sensors (HIH-4000-004, Honeywell) monitored the humidity of the aerosol flow before and after the injection of the humid flow. The amount of water vapor during the counting efficiency measurements was kept constant. Diffusional particle losses were kept to a minimum by keeping tubing as short as possible. In every measurement sequence, particle tubing to CPC and FCE was kept to the same length. In addition, the flow rates were kept constant and the flow was symmetrically split. The additional particle losses for the neutral experiments were calculated following Tauber et al. (2019).

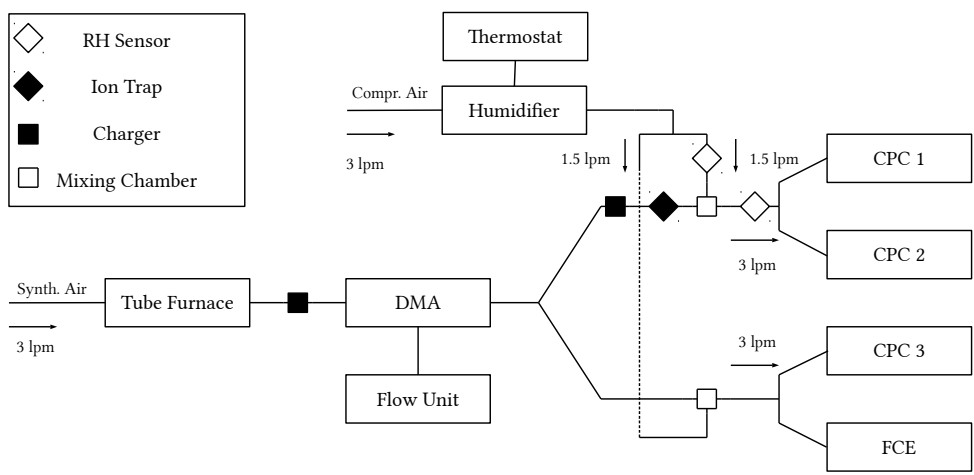

**Figure 1.** The experimental setup for evaluating the relative humidity dependent counting efficiency of continuous flow type CPCs (TSI 3776 UCPC), which was measured relative to a Faraday Cup Electrometer (FCE) (Wlasits, 2019).

Measurements of heterogeneous nucleation of n-butanol at nucleation temperatures ranging from 270 K to 292 K were carried out with the SANC (Wagner et al., 2003). To this end, monodisperse particle fractions were mixed with a carrier gas flow of 5 L/min containing n-butanol vapor and led into the expansion chamber of the SANC. N-butanol vapor was added to the system by controlled injection using a syringe pump, followed by quantitative evaporation of the liquid beam in a heating unit
(Winkler et al., 2008a). As a result, the well-defined and nearly saturated binary vapor-air mixture together with size selected, neutralized monodisperse seed particles from the nDMA was passed into the temperature controlled expansion chamber. Vapor supersaturation was achieved by adiabatic expansion, and the number concentration of droplets nucleating on the seeds was measured with the Constant Angle Mie Scattering (CAMS) method (Wagner, 1985). Based on the CAMS-method a one to one correlation between the time dependent measured scattered light flux under a constant angle and the calculated scattered light
flux as a function of droplet size can be established. Thus, the radius and the number concentration of the growing droplets could be determined simultaneously. This allows us to determine the growth curve of the growing droplets which can be compared to theoretical calculations from a condensation model. Thereby, the saturation ratio can be verified with an accuracy of 2-3 percent. By varying the chamber temperature and the pressure drop in the expansion chamber, different nucleation conditions were analyzed. The nucleation or activation probabilities were measured with the SANC/CAMS method (Wagner et al., 2003).
To evaluate the onset saturation ratio, which corresponds to a nucleation probability value of $P = 0.5$, the experimental data were fitted with a two-parameter fit function (Winkler et al., 2016).

In addition to the SANC measurements, the counting efficiencies of three ultrafine continuous flow type CPCs (Model UCPC 3776, TSI Inc., Minneapolis, USA), which temperature settings were changed over a range of 18 degrees, were measured.

**Table 1.** Temperature settings of the three TSI 3776 UCPCs used in this study in parallel.

| settings | low T [ $^\circ$C ] | standard T [ $^\circ$C ] | high T [ $^\circ$C ] |
|---|---|---|---|
| condenser | 1.1 | 10.0 | 18.9 |
| saturator | 30.1 | 39.0 | 47.9 |
| optics | 31.1 | 40.0 | 48.9 |

In contrast to the SANC, the supersaturation is obtained by saturating a laminar flow and subsequently cooling the aerosol together with the saturated flow in the condenser. Both methods lead to heterogeneous nucleation and condensation of n-butanol vapor on the aerosol particles due to a temperature reduction. Thereby the particles grow to a size at which they can be detected optically. To measure the detection efficiency of the TSI UCPCs, the aerosol flow subsequent to size classification was symmetrically split. The aerosol was passed to three CPCs ($N_{CPC,1-3}$), each operating at different temperature settings (see Table 1), and to the Faraday Cup Electrometer (FCE) ($N_{FCE}$), which was operated in parallel. Thereby the counting efficiency was determined relative to a FCE (Model 3068B Aerosol Electrometer, TSI Inc., Minneapolis, USA). The detection efficiency $\eta$ of a UCPC can be determined by comparing the number concentration of the CPC $N_{CPC,1-3}$ and the total number concentration $N_{FCE}$ according to the following relation:

$$\eta = \frac{N_{CPC,1-3}}{N_{FCE}}. \tag{1}$$

In order to calculate the counting efficiency of the neutralized aerosol, we evaluated the charging efficiency and the theoretical penetration efficiency for the FCE and CPC to correct for the neutralization and penetration losses following Tauber et al. (2019). By operating the CPCs at different temperature settings we were able to analyze simultaneously the detection efficiency with varying peak super saturation ratios (Tauber et al., 2019). The particle detection efficiency mainly depends on the activation probability, which is primarily a function of supersaturation and particle diameter (Barmpounis et al., 2018). By conducting detection efficiency measurements the cut-off diameter is determined. The cut-off diameter corresponds to the mobility diameter at which 50% of the particles are counted in a CPC.

## 3 Results and Discussion

### 3.1 Temperature and Humidity Effects

In this chapter we discuss the measurements of the onset saturation ratio of n-butanol depending on nucleation temperature and humidity with the SANC. The heterogeneous nucleation probability ($P$) for different seeds and seed properties in the size range of 2.5 to 10.5 nm mobility diameter was investigated. The heterogeneous nucleation probability represents the number concentration of activated seeds normalized to the total number concentration of the aerosol. It depends on the saturation ratio which is given by the ratio of partial vapor pressure divided by the equilibrium vapor pressure at the corresponding temperature after expansion (nucleation temperature $T_{nuc}$). Measurements were conducted at constant $T_{nuc}$ by varying the vapor

amount and keeping the pressure drop constant. In a recent study we have shown that with reduced nucleation temperatures the onset saturation ratio needed to activate a certain Ag particle size increases (Tauber et al., 2019). This is in line with classic Kelvin predictions (Thomson, 1871), where with decreasing temperature the required equilibrium saturation ratio increases. This behaviour agrees with homogenous nucleation. To investigate the impact of RH on heterogeneous nucleation probability the Ag measurements were complemented by NaCl seeds. The resulting onset saturation ratios depending on the nucleation temperatures are shown in Figure 2. Clearly, the required saturation ratio to activate a certain particle size is always lower for Ag particles compared to NaCl particles. Heterogeneous nucleation of n-butanol vapor on NaCl and Ag aerosol particles shows a remarkably different behavior. An opposite temperature trend for NaCl seeds was identified when compared to the Kelvin prediction, except for the smallest sodium chloride particles with a mobility diameter of 2.5 nm. Such a trend was not found for silver nano-particles.

This finding is consistent with already published results from Schobesberger et al. (2010) for n-propanol, where an opposite temperature trend for NaCl but not for Ag seeds could be observed. To rule out any humidity effects, the sheath air of the nDMA was monitored using a commerical relative humidity sensor during the measurements. Thereby an accumulation of water in the used silica gel dryer of the sheath air loop could be recorded. The water aggregation in the dryer, originating from the compressed air supply, reached values up to $10\%$ RH at $22°C$ room temperature. As a consequence, the carrier gas supplement for the experiment was changed from dried and filtered compressed air to synthetic air. Additional measurements were performed with a relative humidity $< 3.5\%$ at room temperature.

In Figure 3 the experimental values for neutral sodium chloride seeds are shown and listed in Tables S5-S9 of the supporting information for both considered aerosol species. The necessary onset saturation ratio for activating "dry particles" of a certain size is at the same value as for "high" RH values ($< 10\%$) or (with one exception) increased for dry measurements (RH $< 3.5\%$). In other words, the additional small amount of water vapor during the measurements decreases the energy barrier which has to be overcome to activate a sodium chloride nano-particle with n-butanol in the observed size range. This finding suggests that even small contaminations of water vapor can influence the heterogeneous nucleation of butanol, a finding that agrees with already published studies for n-propanol-water mixtures on Ag and NaCl particles (Wagner et al., 2003).

For the heterogeneous nucleation of n-butanol on silver seeds no decrease of the onset saturation ratio due to the presence of water was observed. The opposite temperature dependence for NaCl and synthetic air as carrier gas (blue symbols in Figure 3) vanishes for small mobility diameters ($< 3.5$ nm), but particle sizes above 3.5 nm still follow the unusal temperature dependence. This indicates that sodium chloride particles act differently in the observed size range. It seems that NaCl seeds at sizes below $\sim 3.5$ nm are less prone to dissolution or restructuring effects. As a result, the nucleation of n-butanol on small NaCl seeds is comparable to silver clusters. The opposite temperature trend vanishes and follows the Kelvin relation. Another remaining question is: Does the charge sign have an effect on heterogeneous nucleation? To answer this question, additional measurements focusing on charge and humidity for NaCl particles in the corresponding nanometer range were conducted.

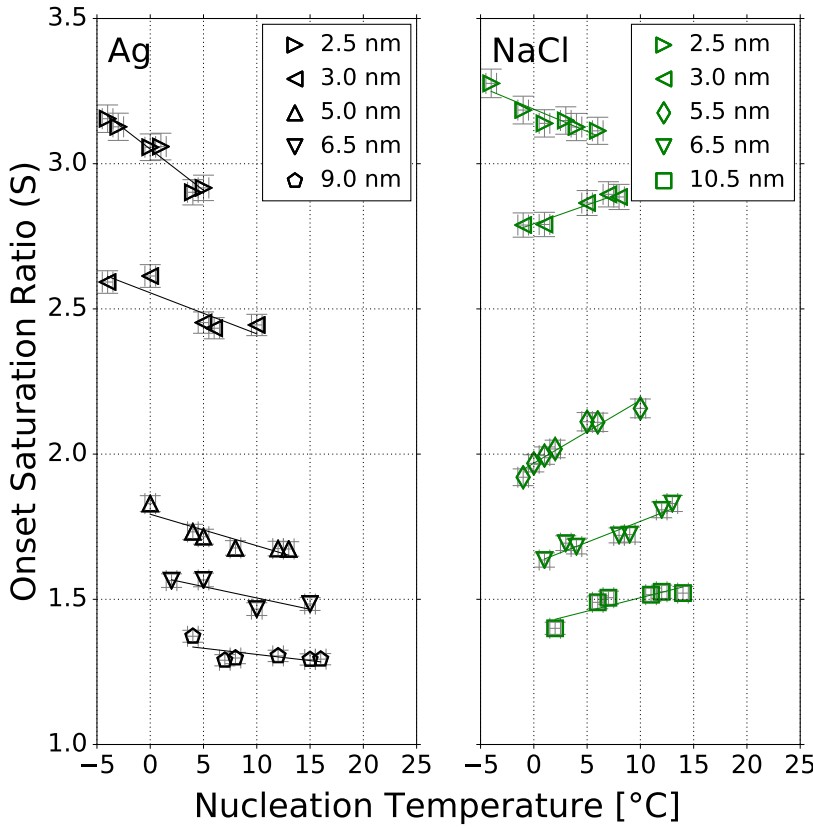

**Figure 2.** The onset saturation ratio versus nucleation temperature for neutral Ag (black) and NaCl (green) particles of different mobility equivalent diameters. With decreasing temperature the onset saturation ratio decreases for sodium chloride and increases for silver nanometer sized particles. Only the 2.5 nm NaCl seeds do not follow the opposite temperature trend. Lines represent a linear fit of data to show the temperature dependence for NaCl and Ag seeds. The onset saturation ratio increases with decreasing seed size for both test aerosols.

### 3.2 The Influence of Charge and Particle Restructuring Effects

To investigate the charge effect on heterogeneous nucleation of n-butanol vapor on silver and sodium chloride particles the neutralizer and ion-trap in front of the expansion chamber (see Figure S1) were removed. Thereby the seeds were positively or negatively charged, depending on the applied voltage to the nDMA. Nucleation studies were conducted with constant nucleation temperatures but varying charge state of the aerosol using synthetic air as carrier gas. For positively charged silver particles no charge dependence on heterogeneous nucleation was found. However, NaCl seeds show a charge dependence, especially below 5 nm. As listed in Table 2 the necessary onset saturation ratio for neutral and positively charged silver particles is below the NaCl values, and no charge enhancement was found. Negatively charged Ag particles could not be fully

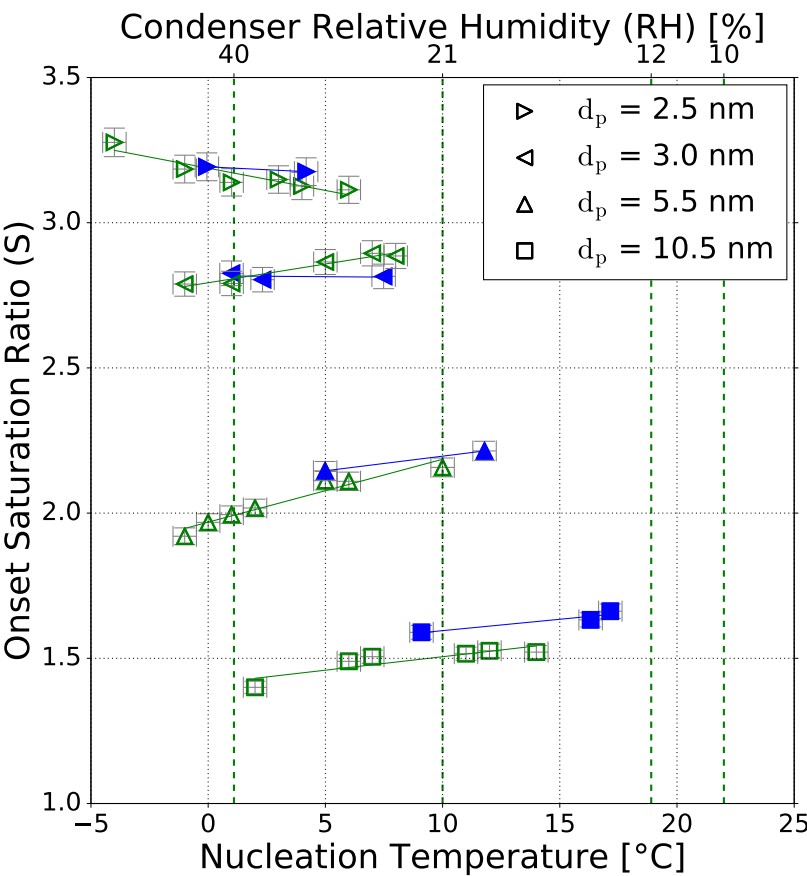

**Figure 3.** The onset saturation ratio versus nucleation temperature for NaCl particles of different mobility euqivalent diameters. The green symbols show the onset saturation values recorded with relative humidity values $\leq 10\%$ and the blue symbols represent measurement values when the RH was $< 3.5\%$. The green dashed vertical lines represent the calculated relative humidities at the corresponding preset condenser temperature (see Table 1) with a starting value of $10\%$ RH at $22°C$. The increased relative humidity inside the condenser of a TSI 3776 UCPC for reduced nucleation temperatures is shown and will be discussed in the counting efficiency measurements section.

investigated due to technical problems with the SANC at the end of the measurement run. Kangasluoma et al. (2016) conducted measurements using tungsten oxide, ammonium sulfate and tetraheptylammonium bromide nano clusters of different charge states to determine the counting efficiency. During their studies a charge enhanced particle activation was observed similar to our findings regarding the NaCl measurements.

5      It is known from prior research that NaCl aerosol particles, which are generated by evaporation and condensation in a tube furnace, undergo a structural change in the presence of water (Krämer et al., 2000; Biskos et al., 2006) or n-propanol (Petersen et al., 2001; Kulmala et al., 2001). As a consequence, the particles shrink in the presence of polar vapors. In order to test a humidity induced change in particle size, we used a standard DMPS setup, which has a filtered and dried (using Silica gel)

**Table 2.** SANC experimental results for the onset saturation ratio for neutral (n), positively (+) and negatively (-) charged particles. $D_p$ is the mean mobility equivalent diameter, $\sigma_g$ the mean geometric standard deviation and $S_{onset}$ the onset saturation ratio.

| $D_p$ [nm] | $\sigma_g$ | Ag ($RH < 2.5\%$) | | | NaCl ($RH < 2.5\%$) | | |
|---|---|---|---|---|---|---|---|
| | | $S_{onset}$ n | $S_{onset}$ + | $S_{onset}$ - | $S_{onset}$ n | $S_{onset}$ + | $S_{onset}$ - |
| 2.5 | 1.073 | 3.10 | 3.07 | - | 3.19 | 3.20 | 3.04 |
| 3.0 | 1.070 | 2.48 | 2.48 | - | 2.81 | 2.72 | 2.64 |
| 3.5 | 1.067 | 2.33 | 2.33 | - | 2.60 | 2.54 | 2.47 |
| 5.5 | 1.065 | 1.85 | 1.85 | 1.83 | 2.21 | - | 2.20 |
| 9.0 | 1.068 | 1.34 | 1.36 | - | - | - | - |
| 10.5 | 1.067 | - | - | - | 1.64 | 1.66 | 1.63 |

sheath air loop. The corresponding setup schema can be found in the supporting information (Figure S3). Our results confirmed the measurements conducted by Biskos et al. (2006), showing the shrinkage of sodium chloride particles in the presence of water vapor. This shrinkage was also investigated in our study for n-butanol. The experimental results for water are shown in Figure 4. We measured monodisperse NaCl aerosols with mobility diameters between 4.5 and 11 nm at relative humdities up to 50%. The shrinkage measurements in Figure 4 were conducted with SHT75 sensors, with an accuracy of $\pm 1.8\%$ RH and with a HMI38 Humidity Data Processor with a HMP35E probe, with an accuracy of $\pm 2.0\%$ RH. However, with decreasing mobility diameter, also the shrinkage becomes less. For example, an 11 nm NaCl particle exposed to 40% RH is shrinking by about 15%. A 4.5 nm particle at 40% RH shrinks only by about 3%. Hence, the structural rearrangment of particles below 5 nm is not as pronounced as for larger ones. This decrease in size for particles $> 5$ nm is a result of solvation on the surface of the NaCl particle caused by the presence of water vapor. Thereby a wet surface accrues (Castarède and Thomson, 2018). Due to this dissociation process and the resulting liquid surface the particle attracts more polar vapor molecules. In addition, the overall polarity of the working fluid of n-butanol and water mixture increases, due to the higher dipole moment of water ($6.2 \times 10^{-30}$ Cm) compared to n-butanol ($5.8 \times 10^{-30}$ Cm) (Reichardt and Welton, 2011). As a result, additional vapor molecules can more readily condense onto the seed aerosol. This mechanism could explain the resulting lower onset saturation ratio needed for NaCl seeds $\geq 3$ nm as shown in Figure 3 for lower nucleation temperatures. Consequently, by reducing the nucleation temperature the saturation vapor pressure decreases, leading to an increase in the saturation ratio for water (RH is increasing, see Figure 3, upper x-axis).

In summary, the SANC measurements allowed us to study the heterogeneous nucleation of n-butanol depending on the charge state, nucleation temperature and humidity of the carrier gas. Charge enhanced nucleation could be found for sodium chloride particles $\leq 3.5$ nm. By taking special care of the water vapor amount a humidity enhanced nucleation for sodium chloride seeds was found. Under dry conditions the observed opposite temperature trend for NaCl seeds $< 3.5$ nm vanishes. For larger sodium chloride particles this trend is found to be persistent.

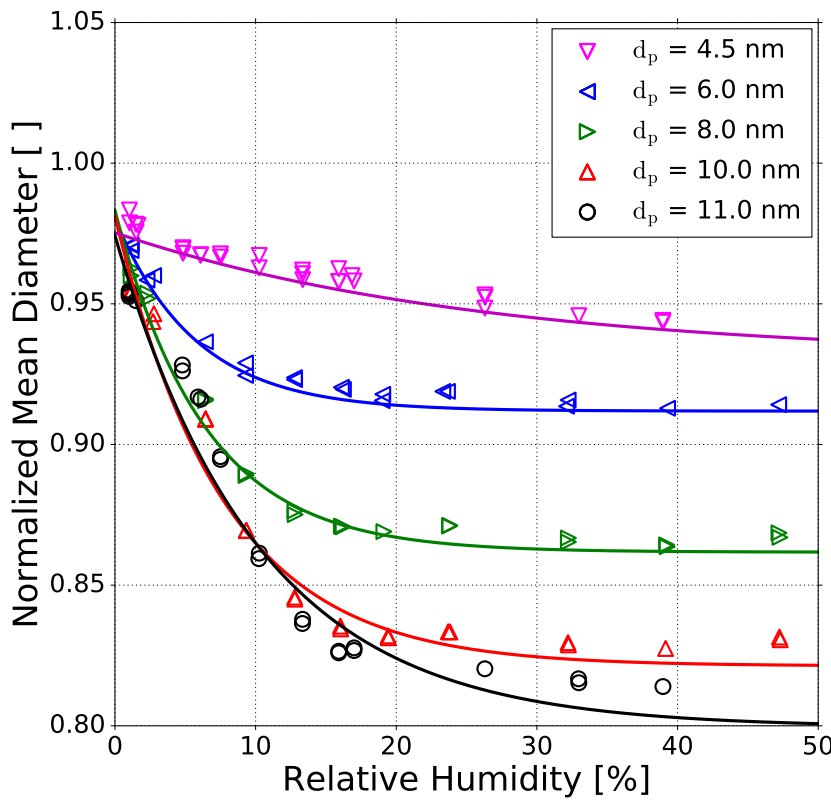

**Figure 4.** Normalized mean diameter of sodium chloride particles as a function of relative humidity which represents the shrinkage of NaCl seeds in the presence of water vapor. All sizes are mobility equivalent diameters. Each line represents a tentative three-parameter fit of an exponential decay. The measurements were conducted with SHT75 sensors with an accuracy of $\pm 1.8\%$ RH and with a HMP35E probe with an accuracy of $\pm 2.0\%$ RH.

### 3.3 Counting Efficiency Measurements

Complementing the SANC measurements we have performed counting efficiency measurements at different condenser temperatures, but constant $\Delta T$ between saturator and condenser for a commerical n-butanol TSI 3776 UCPC. Thereby, we were able to increase or decrease the saturation ratio profile as described by Tauber et al. (2019). The onset saturation ratio and nucleation temperature measured with the SANC for a classified monodisperse seed particle size could be compared to the three different temperature settings of the condenser. The maximum value for the calculated saturation ratio profiles and the extrapolated $S_{onset}$ for NaCl (partly extrapolated from the lines in Figure 2 and 3) under different humidity conditions at low, standard and high condenser temperatures are listed in Table 3. Thus, the ability to activate sodium chloride seeds in the cut-off

**Table 3.** Calculated maxima of the CPC saturation ratio profiles $S_{max}$ for low, standard and high temperature settings (Tauber et al., 2019), in comparision to the extrapolated $S_{onset}$ as measured by the SANC.

| $d_p$ [nm] | | NACL neutral ($RH < 3.5\%$) | | | | NACL neutral ($RH < 10.0\%$) | | | |
|---|---|---|---|---|---|---|---|---|---|
| | | 2.5 | 3.0 | 3.5 | 5.5 | 2.5 | 3.0 | 3.5 | 5.5 |
| T settings | $S_{max}$ | $S_{onset}$ | $S_{onset}$ | $S_{onset}$ | $S_{onset}$ | $S_{onset}$ | $S_{onset}$ | $S_{onset}$ | $S_{onset}$ |
| low | 4.3 | 3.18 | 2.81 | 2.65 | 2.10 | 3.17 | 2.80 | - | 1.99 |
| standard | 3.4 | 3.15 | 2.81 | 2.57 | 2.20 | 3.03 | 2.92 | - | 2.18 |
| high | 2.8 | 3.11 | 2.80 | 2.50 | 2.28 | 2.90 | 3.03 | - | 2.37 |

range of a commerical UCPC could be analyzed. With increasing temperature the maximum saturation ratio ($S$) decreases. As a result, the detection efficiency is shifting to larger particle sizes.

For the verification of our findings we measured the cut-off diameter, charge and relative humditiy dependence for the TSI 3776 UCPC at different temperature settings. By varying the temperature settings of the UCPC we were also able to measure under different condenser (nucleation) temperatures (see Table 1). For comparison, particle classification with the nDMA was always performed under dry conditions (RH < 3.5%), so that the structural change of the sodium chloride clusters resulting from water vapor are kept to a minimum. Subsequent to the size selection, the particles were mixed with humidified air to reach RH values ranging from 0 to 40% as shown in Figure 1. Hence, we were able to vary the relative humidity with an accuracy of ±3.5% by measuring the humidity before and after the mixing zone. In total, about 400 detection efficiency curves were recorded. To rule out any dependence of the individual activation behavior of different CPCs, measurements involving one single CPC set to all three temperature settings one after another, and measurements involving three CPCs that were all set to one specific setting, have been performed. Based on these measurements, the maximum error of the cut-off diameter linked to the individual activation behavior was calculated to be at the highest at 8.1%. Results of earlier studies indicate a discrepancy of the humidity dependence for sodium chloride and silver nano-particles on the detection efficiency of commerical CPCs (Sem, 2002). In the study conducted by Sem (2002), at increased RH the used CPCs had a lower cut-off diameter above 5 nm. In this size range the particle shrinkage plays an important role: The already mentioned emerging liquid surface would attract further vapor molecules. Consequently, the NaCl particles can be easier activated than an insoluble seed.

However, in this study a TSI 3776 UCPC was used, which has a cut-off diameter well below 5 nm. An exemplary detection efficiency measurement for the three CPC temperature settings is depicted in Figure 5. These measurements were conducted under elevated humidity conditions (RH = 10% at room temperature), which lead to a decrease compared to dry conditions (RH = 3.5%) of the necessary onset saturation ratio for neutral NaCl seeds as shown in Figure 3. Due to the temperature decrease in the condenser, the saturation ratio for water vapor increases up to $S = 1.6$ with an inlet RH of 40% at room temperature. Studies conducted by Petersen et al. (2001) on binary heterogeneous nucleation of water-n-propanol vapor mixtures onto NaCl seeds also show that small amounts of water reduce the propanol activity substantially. Butanol is hardly miscible with water in macroscopic mixtures, but still small amounts support the nucleation. The resulting cut-off diameters for neutral, negatively and positively charged sodium chloride nano-particles are shown in Figure 6. It can be clearly seen that with increasing relative

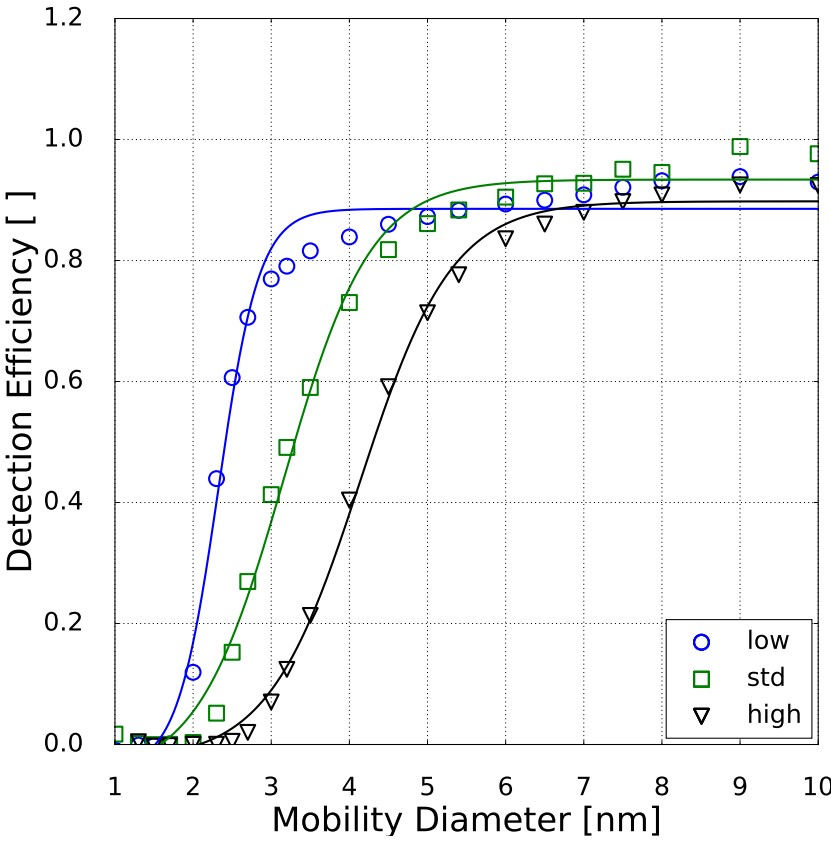

**Figure 5.** Detection efficiency as a function of mobility diameter for neutral sodium chloride particles. The symbols represent the measured counting efficiencies for low (circles), standard (std, squares) and high temperature (triangles) settings at 10% RH at room temperature of the inlet line. The line displays a fit to the measured counting efficiencies (Wiedensohler et al., 1997). Thereby the corresponding cut-off diameter with the detection efficiency of 0.5 can be evaluated.

humidity the cut-off diameter is decreasing for a UCPC operating at standard conditions (green) from about 3.7 to about 2.0 nm for neutral NaCl clusters. To investigate a possible effect of the charge history of neutral particles, positively and negatively charged particles were neutralized. An effect from charge history could possibly be that previously positively charged particles differ from those that were negatively charged prior to neutralization, e.g. by their original properties or because of a change

5 of properties during the neutralization process (Kangasluoma et al., 2013; Alonso and Alguacil, 2008). The resulting cut-off diameters are shown in Figure 6 (left). Regarding the charge history, no difference in activation could be found for sodium chloride. Only under high n-butanol saturation ratios (low temperature settings) and high relative humidity conditions the previously negatively charged particles exhibit a lower cut-off diameter. In Figure 6 (right panel), the cut-off diameters as a function of RH are shown for particles of different polarity. Remarkably, for charged NaCl particles we find increased cut-

off diameters compared to the neutral particles. Since the flow carrying monodisperse aerosols and the flow of humid air are joined together in front of the CPC inlet, the increase of cut-off diameters could be a result of the increased attraction of water molecules to charged NaCl particles under sub-saturated conditions. Thereby the $Na^+$ and $Cl^-$ ions get separated and form a solvent cage around themselves (Castarède and Thomson, 2018). This solvent cage reduces the tendency of ions to aggregate
(Loudon, 2009).

Li and Christopher J. Hogan (2017) performed measurements on the uptake of organic vapor molecules by nanometer scaled sodium chloride cluster ions, using a DMA coupled to a Time-of-Flight (TOF) mass spectrometer. The results show that the polarity and the molecular structure of the vapor molecules control the vapor uptake. This is in agreement with our findings: Increasing amounts of water molecules are mixed with already preexisting butanol molecules. As a result, the overall bulk
polarity of the working fluid increases and the activation of sodium chloride clusters is improved. In contrast to the SANC measurements, no significant sign preference for charge enhanced nucleation could be found. This is probably due to the cut-off diameter size which is for most of the counters $> 3$ nm. Except for negatively charged NaCl particles, which show a charge enhancement as it was oberserved with the SANC for particles $< 3$ nm mobility diameter. In addition, the recorded cut-off diameter for high T settings is in the size range, where the opposite temperature trend was recorded by the SANC
measurements. Consequently, by decreasing the temperature the needed saturation ratio to activate a NaCl particle in this size range is decreasing. Futhermore, the increasing RH supports the structural rearrangement which reduces the cut-off diameter (see Figure 6) compared to dry conditions. As stated in Castarède and Thomson (2018), the solvation of sodium chloride clusters leads to an electrical potential. In this study, we show that nucleation of n-butanol vapor onto soluble NaCl below 3.5 nm is enhanced. This effect takes place as the aerosol is exposed to RH, i.e. before the sample enters the CPC. Subsequently,
inside the CPC, the RH increases due to the temperature reduction inside the condenser which will have an effect on the charge enhancement. As a matter of fact, our results lead to an improved detection efficiency for sodium chloride clusters. At high relative humidity and low temperature settings the detection efficiency is further promoted by charge enhanced nucleation.

Results on the cut-off diameter depending on the relative humidity at the UCPC inlet for neutralized and charged silver seeds are shown in Figure 7. For Ag particles no shrinkage in mobility diameter could be measured, and the clusters at a certain size
can be assumed to be spherical (Winkler et al., 2016). As we can see in Figure 7, no relative humidity dependence on the cut-off diameter within the uncertainty range was observed. However, at reduced supersaturation (black) the cut-off diameter is lower for charged Ag nanoparticles. This indicates a charge enhanced nucleation for silver aerosol particles in the measured size range. Under highly supersaturated conditions (blue) and also at standard settings (green) the charge enhanced nucleation can be neglected. Apparently, the particle charge only supports the nucleation and growth process under high n-butanol and
water (black) saturation vapor pressure conditions i.e. low supersaturation conditions (high CPC Temp settings, see Table 1). In fact, this finding supports the study of Winkler et al. (2008b). They proposed that the needed onset saturation ratio for charged particles is lower than for neutral particles. Under high CPC temperature conditions (high T settings), the saturation ratio is the lowest. The nucleation barrier which has to be overcome to activate charged seeds at the size of the cut-off diameter is therefore lowered by charge enhancement. As a result, the charged silver particles can be activated but not the neutral Ag seeds under
low saturation ratio conditions (black). According to the SANC measurements, no charge enhanced nucleation for positively

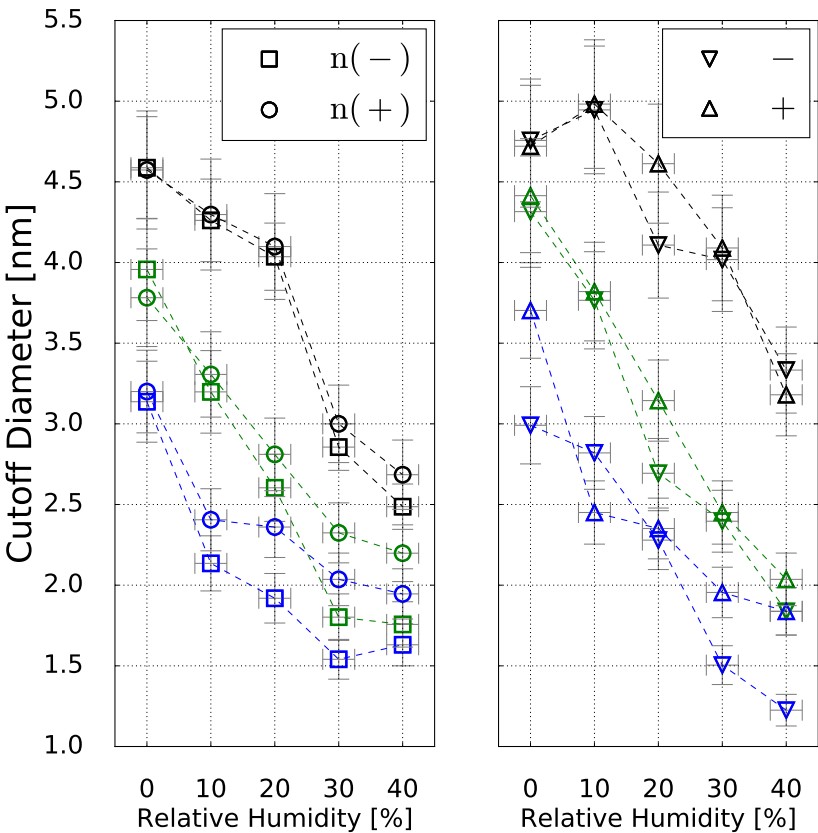

**Figure 6.** Cut-off diameter as a function of RH for neutral (left) and charged (right) sodium chloride seeds for low (blue), standard (green) and high (black) temperature settings.

charged Ag particles could be observed. Similarly, we do not find substantial cut-off diameter reduction for positively charged Ag particles. Only at the lowest temperature settings a sign-preference becomes visible favoring the activation of negatively charged particles.

In general, particle counting at the detection limit of ultra-fine condensation particle counters has to be treated with care. The
5   lowest detectable particle size, i.e. the cut-off diameter of a CPC, is among the largest sources of inaccuracy in the sub-10 nm particle concentration measurement (Kangasluoma et al., 2018). Given that, not only seeds of different chemical compositions but also different particle solubilities are present in the atmosphere – these uncertainties in the determination of the number concentration need to be considered. A straightforward approach to assess the uncertainty of a humid sample flow is to dry the inlet flow of a CPC by e.g. using a Silica gel dryer or Nafion dryer, which is a common method for ambient measurements.
10   Additionally, the inlet RH should be monitored and kept below a threshold value in order to narrow down the uncertainty from

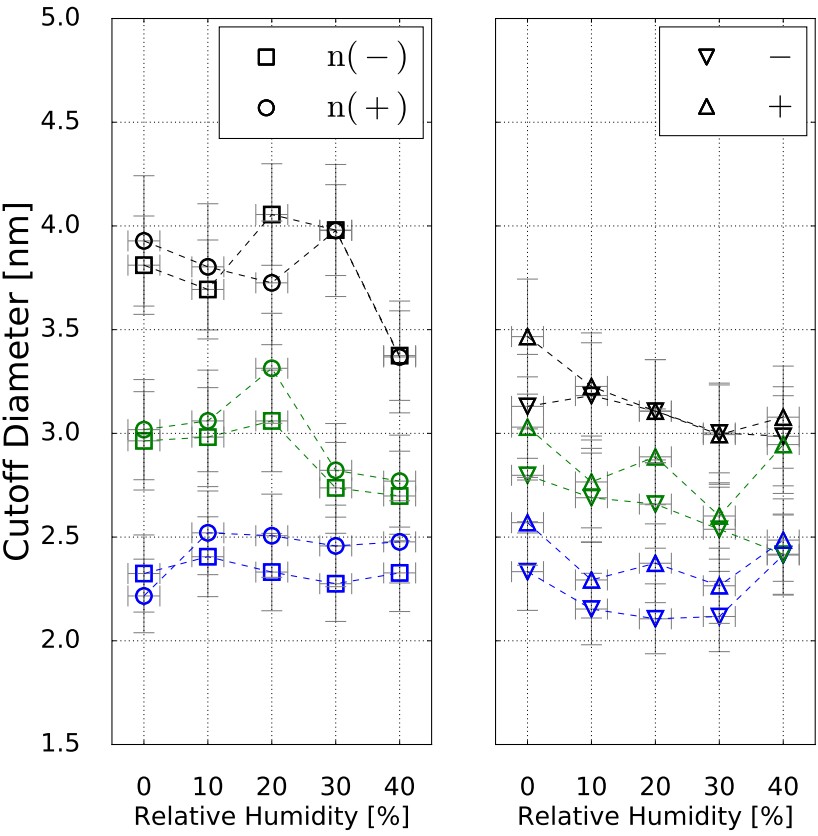

**Figure 7.** Cut-off diameter as a function of RH for neutral (left) and charged (right) Ag seeds for low (blue), standard (green) and high (black) temperature settings.

the humidity dependent change of the cut-off curve in the case of a soluble seed. In the ideal case, measuring instruments are calibrated up to the maximum RH that they are exposed to.

Hygroscopicity measurements of particles in the targeted size range further improve the assessment on the uncertainty of the number concentration. The location of the measurement site is also a clear indicator of a possible enhanced contribution of hygroscopic seed particles, i.e. when the measurement site is located close to the coastline. Thereby the accuracy of the data interpretation would increase. Even without these additional measurements, the location of the measurement site and the source region of the seed particles can be used for data interpretation. As a result, studies that consider sea spray as a particle source should consider the effect of humidity on the particle detection efficiency. Furthermore, studies that investigate aerosol from sources that involve soluble seeds, should consider the effect of humidity on the particle detection efficiency. Soluble seeds do not only include sodium chloride particles which were analyzed in this study. Ammonium sulfate, which originates from the gas-phase from precursor gases having both natural and anthropogenic origin (Sakurai et al., 2005; Smith et al., 2004), can act

as soluble seed. Consequently, we suggest to consider a possible shift in cut-off diameter when such soluble species are present in the sample.

Based on solubility effects and the increased polarity of the working fluid due to the addition of water molecules, similar effects using different working fluids can be assessed by interpreting their chemistry. Diethylene glycol (DEG) is a well polarized molecule due to three oxygen atoms per DEG molecule. Adding water does not increase the overall polarity as much as in the case of butanol. NaCl particles are soluble in DEG as well. As presented in Kangasluoma et al. (2013) variation of the inlet flow RH leads to increased detection efficiency for a turbulent mixing type CPC.

Working fluids like perfluorcarbonates (Fluorinert, FC-43) consist of carbon chains with fluorine atoms instead of hydrogen atoms. These molecules have a low dipole moment. Due to the spherical structure and highly electronegative atoms on the outside the overall surface of the molecule is slightly negatively polarized. Consequently, a shift in the cut-off diameter can be expected by using a mixture of perfluorcarbonates and water. The overall polarity of the working fluid would almost exclusively be based on the dipole moment of water molecules. Pulse height analysis measurements by Hanson et al. (2002), revealed a pronounced substance dependency and a difference in peak distribution when using n-butanol or FC-43 as working fluid in a CPC. The interaction between NaCl seed and water molecule remains regardless of the working fluid. If RH cannot be kept at 0%, the instrument should be calibrated to the corresponding conditions and a possible change in cut-off diameter be considered in the data analysis to ensure high data quality. Especially at measurement sites located near the sea or at changing/high RH, which is the case for most regions.

## 4   Conclusions

The presented measurements conducted with expansion and continuous flow type CPCs have shown that sodium chloride particles with a mobility diameter below 10 nm indicate different activation regimes. According to the SANC measurements, the required saturation ratio to activate a certain particle size is always lower for Ag particles compared to NaCl particles. As a consequence, the cut-off diameters measured for NaCl particles under dry conditions are always higher than for Ag seeds in butanol-based CPCs. Under dry conditions for particles $\leq 3$ nm no opposite temperature trend compared to classical Kelvin predictions was observed. However, for negatively charged NaCl seeds a charge enhanced activation was measured with the SANC. Above 3.5 nm the charge does not play a role during the nucleation process. Charge enhanced particle activation did only occur at the lowest CPC temperature settings. We conclude that the activation of NaCl particles can be enhanced by increased humidity. For Ag this humidity dependence could not be observed. This is an indicator for the importance of molecular interactions between seed and vapor molecules.

A recently published study on heterogeneous nucleation of n-butanol on monoatomic ions has shown that the initial forma-tion of a critical cluster does not favor one of the two possible polarities of the seed particle (Tauber et al., 2018). However, under sub-saturated conditions the existence of seed particle charge plays and important role concerning vapor uptake. Hence the vapor molecules are in contact with the NaCl particles under sub-saturated conditions before the temperature drops and can attach to the seeds. We performed measurements with NaCl seeds which are known to be hygroscopic. Therefore the water

molecules dissociate the NaCl cluster and a solvation process takes place. As a consequence, the charged sodium chloride seeds attract the vapor molecules stronger and thereby the nucleation and condensational growth can start at smaller seed sizes. In this work we assumed that the chemical composition and charging state are the driving forces of nucleation and growth in the size range of the cut-off diameter of the used UCPC. Hence we recommend that in future studies the chemical composition

of the generated particles in different charging states should be investigated.

As described by Li and Christopher J. Hogan (2017), the impact of vapor polarity in the measured particle size range, where interactions between single atoms are of importance, cannot be neglected. Additional water molecules increase the dipole moment of the n-butanol/water mixture and can change the structure of the sodium chloride cluster. If water vapor is present charge effects on the surface promote the initial steps of nucleation even more. As a result, the energy barrier for

activating nanometer sized NaCl particles can be reduced - this is comparable to already published results for n-propanol/water mixtures conducted by Petersen et al. (2001). Our data shows a strong effect of RH on heterogeneous nucleation of n-butanol on NaCl particles, but not for Ag seeds. This finding suggests that the different RH sensitivity is less a consequence of binary heterogeneous nucleation, but rather attributable to changing seed properties in the interaction with vapor molecules. According to the NaCl shrinkage measurements, a change in diameter during the presence of water vapor - which we associated with a

restructural effect - could be observed.

To improve our understanding of the observed restructuring effect, future studies on heterogeneous nucleation depending on different seeds and humidity should be conducted. Also the herein presented measurements could be extended with the size selection after the particle restructuring. Additionally, higher RH conditions could be used to investigate the restructural effect on even smaller particles. In particular, the humidity dependent activation of hygroscopic nanoparticles which are present in

the atmosphere would be of great interest. Thereby, the molecular interactions between the seed and the vapor molecules can be further evaluated which may lead to a model description.

Our results suggest that atmospheric measurements of ambient nanoparticles should take into account possible RH effects on the instrument's cut-off diameter, when a large hygroscopic particle fraction is present. Thus, number concentration measurements should be paired with chemical and/or hygroscopicity measurements to improve the assessment of the uncertainty.

If these additional measurements cannot be conducted, the location of the measurement site should be taken into account for data interpretation. Especially when conducting studies in marine surroundings, at which sea spray is one of the contributing particle sources, special care should be taken during the data evaluation (Lawler et al., 2014; Zieger et al., 2017). As a standard procedure, in many atmospheric studies the CPC inlet flow is dried before entering the instrument. We suggest that additional monitoring of the RH of the inlet flow is critical, since the variation in RH between 0 and 40 % even shows a pronounced

shift in the cut-off diameter. This would lead to an overestimation in the number concentration of nucleation mode particles. In ambient conditions this mode usually contains high number concentrations of particles. Comparison between measurements taken in different regions with varying hygroscopic nanoparticle concentrations thus need to be treated with care to avoid wrong interpretation.

*Data availability.* Supplementary data associated with this article can be found in the online version.

*Author contributions.* Christian Tauber designed the setup, Christian Tauber performed the SANC experiments, Christian Tauber, Sophia Brilke and Peter Josef Wlasits performed the detection efficiency measurements, Paulus Salomon Bauer and Gerald Köberl performed the shrinkage measurements, Christian Tauber, Sophia Brilke, Peter Josef Wlasits, Gerhard Steiner and Paul Martin Winkler were involved in the scientific interpretation and discussion, and Christian Tauber, Sophia Brilke, Peter Josef Wlasits and Paul Martin Winkler wrote the manuscript.

*Competing interests.* The authors declare that they have no conflict of interest.

*Acknowledgements.* This work was supported by the European Research Council under the European Community's Seventh Framework Programme (FP7/2007/2013)/ERC grant agreement no. 616075. Dimitri Castarède and Lennart Graewe are thanked for helpful discussions.

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
