# Peer review of "Humidity effects on the detection of soluble and insoluble nanoparticles in butanol operated condensation particle counters"

_Atmospheric Measurement Techniques, 2019_

## Referee Comment (RC1) · Anonymous Referee #1 · 1 Mar 2019

There are some interesting results but the presentation could use some work. Primarily, the humidity effect for detection of NaCl nanoparticles is prominently displayed in a couple of figures but this effect is nearly lost because of all the text about charge or charging effects. The authors need to motivate, and consider significantly paring down, the data and discussion of charge effects.

Also important is that there is a lack of discussion on how others could apply the humidity effect for these types of particles.

A concerning factor as to whether these results apply to the atmosphere: Are there actually any nanometer-sized NaCl aerosol in the atmosphere?

[Figure]

The discussion of the rearrangement / change in shape of these particles is missing any experimental description. This experimental data is a (another?) humidity effect and seems to be pertinent to the topic here.

The discussion around this shape effect would be greatly helped if data could be presented with the DMA run at the RH of interest. Why was not a recirculating pump / filter not used for some measurements? That would help the interpretation of the overall results and may lead to some sort of basis for a model description.

There is previous work on humidity effects for nanometer-sized sulfuric acid particles out of the Eisele-McMurry collaboration at NCAR. Also the O'Dowd group explored chemical effects and the activation of nano-particles. Those two studies were focused on pulse height analysis but the humidity effect explored here is integral to those experiments. I think the Donaldson group out of Toronto also discussed humidity effects in butanol CPCs. There are probably others and they should be cited. How would the PSM (Seinfeld, Kulmala etc.) instrument data be affected? How about alternate condensing fluids like diethylene glycol or FC43 (the latter used on aircraft campaigns by Brock and co-workers) ?

The authors should consider changing the language from inverse temperature to something like non-congruent (with the Kelvin ....) temperature dependence. Inverse temperature to many physical scientist means a 1/T dependence....

The non-native English speaking authors have had difficulty translating their thoughts into English. Hard to follow their logic at times. More on this at a later time: wanted to get these major comments to the authors for discussion etc.
* * *

---

## Referee Comment (RC2) · Anonymous Referee #2 · 4 Mar 2019

Summary: The paper demonstrates a humidity dependence on NaCl activation using a butanol CPC, where increasing humidity decreases the activation cut-size, while showing no such dependence with Ag particles. The measurement was performed with both continuous flow CPCs and the SANC. Due to the strong humidity-dependence of some particles, it is believed that ambient measurements could be activated at smaller sizes than their laboratory-controlled equivalent and care should be taken in assuming constant cut-sizes for butanol-based CPCs in ambient measurements of unknown composition.

These results seem significant and interesting, but I think it could be reorganized to

better integrate, introduce, and motivate each experiment's contribution to the objective. Some experiments are not fully motivated of their potential significance until the results section.

The abstract/intro is missing a significant section of the paper regarding charge effect. Poor humidity sensor selection for a paper on targeting humidity effects. Sensor uncertainties need to propagate through to resulting figures (e.g. Figure 4).

Would a working fluid such as Fluorinert counteract the humidity dependence of NaCl activation, or would the NaCl still uptake water and nucleate more easily with humidity? Practical solutions and guidelines for scientists would be helpful.

Major Comments: The citations provided, on nucleation for example (P2 L5), should credit the work from previous authors. Also, heterogeneous nucleation mechanisms needs to include more sources than Wang et al. 2013. In general, the sources need to be expanded to include the major pioneers of the subjects. If the experiments were already performed with propanol in Schosbesberger et al., why would a different behavior with butanol be expected? The difference between the papers, aside from changing the working fluid, should be highlighted.

Figure 1 does not include control system mechanism for RH. The paper focuses on humidity as a primary variable, but not much detail was provided on how it was varied and controlled. Is RH controlled with a feedback loop to account for transients throughout the experiment? If RH is the main variable of interest, its introduction to the system and control should not be glossed over.

Honeywell sensors mentioned are crude (accuracy +/- 3.5%, hysteresis 3%, repeatability +/- 0.5%, etc.). These uncertainties need to be reflected in the figures.

Regarding the counting efficiency experiment (P4), each TSI CPC can have a unique counting efficiency based on laser age, optics cleanliness, alignment, etc. Characterizing the effect of CPC deltaT on nucleation effectiveness using 3 different CPCs does

not seem substantial enough, as each CPC could have a different counting efficiency at the same temperature. It would have been preferable to compare one CPC and repeat the experiment 3 times, so the additional factors mentioned above are constrained. Otherwise, it would need to be stated that the 3 CPCs have been normalized to one another for each of the deltaTs (or proven operation is identical).

The charge effect isn't addressed in the abstract or introduction/background and seems significant to the NaCl activation. The inclusion of this experiment is good, but it feels like it came only at the result of Section 3.1, and didn't have its own proper introduction to motivate why it's included.

Minor Comments: P1 L14-22: This paragraph seems irrelevant to the focus of the paper goals and losing it would not detract from the message. The overview seems a little vague, when it would benefit with a background more targeted to the objective.

Commas must be added in sentences using passive voice. This is done with some, but not all sentences and must be corrected throughout.

Simplify wordy sentences throughout: e.g. P2 L4: "This process is called nucleation and arises..." could simplify to "This process of nucleation arises..."; and P2 L18.

P2 L7: Remove "a" to make CCN plural to agree with your verb and that you used "nuclei" instead of "nucleus", i.e. "...nanometer-sized particles contribute as cloud condensation nuclei (CCN)."

P2 L15-17: You describe deliquescence and efflorescence without using the term for the mechanism.

P2 L 20: Subject-verb agreement; "One technique... is CPCs", not "are".

---

## Author Comment (AC1) · 27 Mar 2019

We appreciate the thoughtful comments by referee 1. For discussion purposes we would like to respond to the general points raised.

*There are some interesting results but the presentation could use some work. Primarily, the humidity effect for detection of NaCl nanoparticles is prominently displayed in a couple of figures but this effect is nearly lost because of all the text about charge or charging effects. The authors need to motivate, and consider significantly paring down the data and discussion of charge effects.*

[Figure]

***Also important is that there is a lack of discussion on how others could apply
the humidity effect for these types of particles.***

We acknowledge the well-justified comment on how others could apply the humidity
effect for these types of particles. Generally, we think that particle counting at the de-
tection limit of ultra-fine condensation particle counters has to be treated with care. Due
to high particle number concentrations in the nucleation mode, atmospheric measure-
ments of soluble and insoluble nanoparticles might lead to wrong data interpretation.
The lowest detectable particle size, i.e. the cut-off diameter of a CPC, is among the
largest sources of inaccuracy in the sub-10 nm particle concentration measurement
(Kangasluoma et al., 2018). Given that not only seeds of different chemical composi-
tions but also solubilities are present in the atmosphere – these uncertainties on the
number concentration need to be considered. A straightforward approach to assess
the uncertainty of a humid sample flow is to dry the inlet flow of a CPC by e.g. using
a Silica gel dryer or Nafion dryer which is a common method during ambient measure-
ments. Additionally, the inlet RH should be monitored and kept below a threshold value
in order to narrow down the uncertainty from the humidity dependent change of the
cut-off curve in the case of a soluble seed.

Hygroscopicity measurements of particles in the targeted size range further improve
the assessment on the uncertainty of the number concentration. The location of the
measurement site is also a clear indicator of a possible enhanced contribution of hy-
groscopic seed particles, i.e. when the measurement site is located by the coastline.
Thereby the accuracy of the data interpretation would increase. Even without these
additional measurements, the location of the measurement site can be used for data
interpretation. As a result, studies that consider sea spray as a particle source should
consider the effect of humidity on the particle detection efficiency. Furthermore, the
enhanced activation of soluble particles can also be used to improve the detection
efficiency for hygroscopic seeds.

***A concerning factor as to whether these results apply to the atmosphere: Are***

*there actually any nanometer-sized NaCl aerosol in the atmosphere?*

The important fact, which will be more highlighted in the updated manuscript, is that in our case NaCl served as a representative hygroscopic seed particle. It was selected for our studies, since it is known to be very hygroscopic and well characterized by different studies like Biskos et al. (2006) and Krämer et al. (2000). Clarke et al. (2003) were able to characterize the size distribution of particles in the atmosphere produced by breaking waves. They could demonstrate that those particles cover the size range from 10 nm to greater than 10 $\mu m$. Regarding the chemical composition of nanoparticles in marine surroundings, M. J. Lawler et al. (2014) conducted TDCIMS measurements in Mace Head, Ireland. The composition measurement of particle between 15-85 nm showed measurable levels of sodium and chloride. Sea spray is a complex mixture of inorganic salts and organic material and a recent study by Zieger et al. (2017) showed that the hygroscopicity of the inorganic component is about 8-15% lower than of pure sodium chloride.

*The discussion of the rearrangement / change in shape of these particles is missing any experimental description. This experimental data is a (another?) humidity effect and seems to be pertinent to the topic here.*

At relative humidity below the deliquescence threshold NaCl particles with an initial mobility diameter between 19-200 nm undergo a microstructural rearrangement as shown by Krämer et al. (2000). Subsequently, G. Biskos et al. (2006) investigated the deliquescence and efflorescence of NaCl seeds with a mobility diameter between 6-60 nm under different humidity conditions using a tandem DMA setup. Both studies revealed a shrinkage with relative humidity conditions below the deliquescence threshold for dry sodium chloride particles. The cut-off diameter of the used TSI 3776 UCPC is at 2.5 nm according to the manufacturer at standard temperature settings. Therefore, we were aiming at studying the size range around the UCPC cut-off diameter using a tandem DMA setup to investigate the shrinkage of NaCl particles when exposed to a defined RH (the setup is also shown in Figure 1 in the comments of the second re-

viewer). Due to the low particle concentrations below 4.5 nm our measurements were limited to this particle size, but still the microstructural rearrangement yielded an attenuated decrease in size with decreasing particle diameter. Therefore, we assumed that the rearrangement or change in shape plays an important role for seed sizes >3 nm, but below the measured charge enhanced nucleation for sodium chloride particles influence the nucleation and growth process (C. Tauber et al., 2018). As a result, water molecules dissociate the NaCl cluster and a charge enhanced nucleation / growth leads to an enhanced activation.

*The discussion around this shape effect would be greatly helped if data could be presented with the DMA run at the RH of interest. Why was not a recirculating pump / filter not used for some measurements? That would help the interpretation of the overall results and may lead to some sort of basis for a model description.*

In order to test a humidity dependent change in particle size, we used a standard DMPS setup which has a filtered and dried (using Silica gel) sheath air loop. Our results confirmed the measurements conducted by Biskos et al. (2006) showing the shrinkage of sodium chloride particles in the presence of water vapor. Nevertheless, in the future we would like to conduct these measurements with humidified sheath air for comparison. Additionally, higher RH conditions could be used to investigate the influence of deliquescence on even smaller particles.

*There is previous work on humidity effects for nanometer-sized sulfuric acid particles out of the Eisele-McMurry collaboration at NCAR. Also the O'Dowd group explored chemical effects and the activation of nano-particles. Those two studies were focused on pulse height analysis but the humidity effect explored here is integral to those experiments. I think the Donaldson group out of Toronto also discussed humidity effects in butanol CPCs. There are probably others and they should be cited. How would the PSM (Seinfeld, Kulmala etc.) instrument data be affected? How about alternate condensing fluids like diethylene glycol or FC43*

***(the latter used on aircraft campaigns by Brock and co-workers)?***

We acknowledge the well-justified comment and thank the reviewer for making us aware of the pulse height analysis studies. The pulse height analysis conducted by D. R. Hanson et al. (2002) for sulfuric acid particles reveals a strong chemical composition and water vapor dependence. The size resolved chemical composition measurements of nanoparticles from reactions of sulfuric acid with ammonia and dimethylamine, investigated by H. Chen et al. (2018), suggest that small, acidic and newly formed particles can affect the physiochemical properties and thereby enhance early particle growth. We recommend that in further studies the chemical composition of the generated particles in different charging states are investigated. We assume that the chemical composition and charging state are the driving forces of nucleation and growth in size range of the cut-off diameter of the used UCPC.

Based on solubility effects and the increased polarity of the working fluid due to the addition of water molecules, similar effects using different working fluids can be assessed by interpreting their chemistry. DEG is a well polarized molecule due to three oxygen atoms per DEG molecule. Adding water does not increase the overall polarity as much as in the case of butanol. NaCl particles are soluble in DEG as well. As a result, it can be inferred that such a shift in the cut-off diameter is expected to be smaller than in the case of butanol. The working principle of PSMs (turbulent mixing) establishes a highly dynamic environment that leads to an increased rate of particle activation even without the presence of water (P. J. Wlasits, 2019).

Perfluorcarbonates (Fluorinert, FC-43): These compounds consist of carbon chains with fluorine atoms instead of hydrogen atoms. These molecules have a low dipole moment. Due to the spherical structure and very electronegative atoms on the outside the overall surface of the molecule is slightly negatively polarized. Consequently, a shift in the cut-off diameter can be expected by using a mixture of Perfluorcarbonates and water. The overall polarity of the working fluid would almost exclusively be based on the dipole moment of water molecules. Also, the pulse height analysis measurements by D. R. Hanson et al. (2002) revealed a weak size change when using FC-43 as

working fluid.

***The authors should consider changing the language from inverse temperature to some-thing like non-congruent (with the Kelvin ....) temperature dependence. Inverse temperature to many physical scientist means a 1/T dependence..***. . .

We thank the reviewer for the comment and we will reformulate to opposite temperature trend.

***The non-native English speaking authors have had difficulty translating their thoughts into English. Hard to follow their logic at times. More on this at a later time: wanted to get these major comments to the authors for discussion etc.***

The manuscript will be revised concerning grammar and word order.

---

## Author Comment (AC2) · 27 Mar 2019

We appreciate the thoughtful comments by referee 2. For discussion purposes we would like to respond to the general points raised.

*Summary: The paper demonstrates a humidity dependence on NaCl activation using a butanol CPC, where increasing humidity decreases the activation cut-size, while showing no such dependence with Ag particles. The measurement was performed with both continuous flow CPCs and the SANC. Due to the strong humidity-dependence of some particles, it is believed that ambient measurements could be activated at smaller sizes than their laboratory-controlled equiv-*

[Figure]

*alent and care should be taken in assuming constant cut-sizes for butanol-based
CPCs in ambient measurements of unknown composition.*

*These results seem significant and interesting, but I think it could be reorganized to better integrate, introduce, and motivate each experiment's contribution
to the objective. Some experiments are not fully motivated of their potential significance until the results section.*

*The abstract/intro is missing a significant section of the paper regarding charge
effect. Poor humidity sensor selection for a paper on targeting humidity effects.
Sensor uncertainties need to propagate through to resulting figures (e.g. Figure
4).*

A paragraph in the introduction for the charge effect will be added in the updated
manuscript. The shrinkage measurements in Figure 4 were conducted with SHT75
sensors, with an accuracy of +/- 1.8% and with a HMI38 Humidity Data Processor
with a HMP35E probe, with an accuracy of +/- 2.0%. For the sake of readability uncertainties were not included in Figure 4 in the final version of the manuscript. The
corresponding uncertainties are discussed in the figure caption instead.

*Would a working fluid such as Fluorinert counteract the humidity dependence
of NaCl activation, or would the NaCl still uptake water and nucleate more easily
with humidity?Practical solutions and guidelines for scientists would be helpful.*

These compounds consist of carbon chains with fluorine atoms instead of hydrogen
atoms. These molecules have a low dipole moment. Due to the spherical structure
and very electronegative atoms on the outside the overall surface of the molecule is
slightly negatively polarized. As a result, a shift in the cut-off diameter can be expected
by using a mixture of Perfluorcarbonates and water. The overall polarity of the working
fluid would almost exclusively be based on the dipole moment of water molecules. In
fact, the interaction between the NaCl seeds and the water molecules will be persistent
if only the working fluid gets changed. To remove the influence the carrier gas should

be kept dry. In case this cannot be done properly, a possible influence of the relative humidity should be kept in mind, especially at measurement sites located near the sea.

*Major Comments: The citations provided, on nucleation for example (P2 L5), should credit the work from previous authors. Also, heterogeneous nucleation mechanisms needs to include more sources than Wang et al. 2013. In general, the sources need to be expanded to include the major pioneers of the subjects. If the experiments were already performed with propanol in Schobesberger et al., why would a different behavior with butanol be expected? The difference between the papers, aside from changing the working fluid, should be highlighted.*

We acknowledge the well-justified comment on the citations of heterogeneous nucleation mechanism and will include them in the updated manuscript. We were not expecting a different behavior. Initially the temperature dependence found by Schobesberger et al. (2010) was used to explain the enhanced detection efficiencies for NaCl found during the laboratory measurements. During the measurements it was found that the relative humidity dependence was stronger than the temperature dependence. Schobesberger et al. (2010) focused on the temperature dependence and compared it to different theoretical approaches like Fletcher theory. In this work we focused on the deviating activation behavior of butanol-based CPCs. The deviating activation behavior is caused by the temperature settings and by increased RH levels.

*Figure 1 does not include control system mechanism for RH. The paper focuses on humidity as a primary variable, but not much detail was provided on how it was varied and controlled. Is RH controlled with a feedback loop to account for transients through-out the experiment? If RH is the main variable of interest, its introduction to the system and control should not be glossed over.*

Figure 1 will be moved into the supplemental materials. In the revised manuscript it will be replaced by the figure shown below. The carrier gas was humidified by passing it through a diffusion type humidifier. The relative humidity of the carrier gas was

[Figure]

monitored and recorded throughout every measurement. The deviation of the recorded values was always below +/- 1% which is below the sensor accuracy.

***Honeywell sensors mentioned are crude (accuracy +/- 3.5%, hysteresis 3%, repeatability +/- 0.5%, etc.). These uncertainties need to be reflected in the figures.***

The humidity measurements were conducted with two different humidity sensors. The cut-off diameter measurements were conducted with the Honeywell sensors. Due to the accuracy of +/- 3.5% the minimal RH difference between two measurements was 10%. The shrinkage measurements were conducted with SHT75 sensors with an accuracy of +/- 1.8%, hysteresis 1%, repeatability +/- 0.1%. In the final version of the manuscript the error bars were not included in Figure 4 for improved readability. Uncertainties will be discussed in the figure caption.

***Regarding the counting efficiency experiment (P4), each TSI CPC can have a unique counting efficiency based on laser age, optics cleanliness, alignment, etc. Characterizing the effect of CPC deltaT on nucleation effectiveness using 3 different CPCs does not seem substantial enough, as each CPC could have a different counting efficiency at the same temperature. It would have been preferable to compare one CPC and repeat the experiment 3 times, so the additional factors mentioned above are constrained. Otherwise, it would need to be stated that the 3 CPCs have been normalized to one another for each of the deltaTs (or proven operation is identical).***

In total about 400 detection efficiency curves were recorded. To rule out any dependence on the individual activation behavior of different CPCs, measurements involving one single CPC set to all three temperature settings one after another, and measurements involving three CPCs that were all set to one specific setting, have been performed. Based on these measurements the maximum error linked to the individual activation behavior was calculated to be at the highest at 8.1% (P. J. Wlasits, 2019). Figure 2 and 3 are chosen as exemplary measurements for different temperature set-
tings and CPCs at the same T settings (C. Tauber et al., 2019).

***The charge effect isn't addressed in the abstract or introduction/background and seems significant to the NaCl activation. The inclusion of this experiment is good, but it feels like it came only at the result of Section 3.1, and didn't have its own proper introduction to motivate why it's included.***

We thank the reviewer for making us aware of the leak in the introduction section. In the final version of the manuscript we will add an additional paragraph on the charge dependence to the introduction.

***Minor Comments: P1 L14-22: This paragraph seems irrelevant to the focus of the paper goals and losing it would not detract from the message. The overview seems a little vague, when it would benefit with a background more targeted to the objective.***

We will remove this paragraph and add an introduction paragraph on the charge dependence / effect.

***Commas must be added in sentences using passive voice. This is done with some, but not all sentences and must be corrected throughout.***

We will review the manuscript to correct that.

***Simplify wordy sentences throughout: e.g. P2 L4: "This process is called nucleation and arises..." could simplify to "This process of nucleation arises..."; and P2 L18.***

This change will be included in the updated manuscript.

***P2 L7: Remove "a" to make CCN plural to agree with your verb and that you used "nuclei" instead of "nucleus", i.e. "...nanometer-sized particles contribute as cloud condensation nuclei (CCN)."***

This change will be included in the updated manuscript.

[Figure]

***P2 L15-17: You describe deliquescence and efflorescence without using the term for the mechanism.***

This will be explained in the updated manuscript.

***P2 L 20: Subject-verb agreement; "One technique...is CPCs", not "are".***

This change will be included in the updated manuscript.
* * *
[Figure]

[Figure]

**Fig. 1.** The experimental setup for evaluating the relative humidity dependent counting efficiency of continuous flow type CPCs (TSI 3776 UCPC), which was measured by operating a Faraday Cup Electrometer (FCE)

**Fig. 2.** NaCl positive with 10% RH, all curves were recorded with the same CPC.

**Fig. 3.** NaCl negative with 0% RH, all curves were recorded with different CPCs but at constant temperature settings.

---

## Author Response (AR1)

**Authors Response to the Reviewer Comments regarding the manuscript "Humidity effects on the detection of soluble and insoluble nanoparticles in butanol operated condensation particle counters" by Tauber et al.**

Dear editor,

we appreciate the thoughtful comments by the two reviewers which have substantially helped to improve and clarify our manuscript. All comments have been considered and are explained in the detailed point-by-point answers.

Sincerely,

Christian Tauber on behalf of all co-authors

**Reviewer: 1**

General comments:

*There are some interesting results but the presentation could use some work. Primarily, the humidity effect for detection of NaCl nanoparticles is prominently displayed in a couple of figures but this effect is nearly lost because of all the text about charge or charging effects. The authors need to motivate, and consider significantly paring down the data and discussion of charge effects.*

*Also important is that there is a lack of discussion on how others could apply the humidity effect for these types of particles.*

**Response:**
We acknowledge the well-justified comment on how others could apply the humidity effect for these types of particles. Generally, we think that particle counting at the detection limit of ultra-fine condensation particle counters has to be treated with care. Due to high particle number concentrations in the nucleation mode, atmospheric measurements of soluble and insoluble nanoparticles might lead to wrong data interpretation. The lowest detectable particle size, i.e. the cut-off diameter of a CPC, is among the largest sources of inaccuracy in the sub-10 nm particle concentration measurement (Kangasluoma et al., 2018). Given that not only seeds of different chemical compositions but also soluble substances are present in the atmosphere, these uncertainties on the number concentration need to be considered. A straightforward approach to assess the uncertainty of a humid sample flow is to dry the inlet flow of a CPC by e.g. using a Silica gel dryer or Nafion dryer which is a common method during ambient measurements. Additionally, the inlet RH should be monitored and kept below a threshold value in order to narrow down the uncertainty from the humidity dependent change of the cut-off curve in the case of a soluble seed.
Hygroscopicity measurements of particles in the targeted size range further improve the assessment on the uncertainty of the number concentration. The location of the measurement site is also a clear indicator of a possible enhanced contribution of hygroscopic seed particles, i.e. when the measurement site is located by the coastline. Thereby the accuracy of the data interpretation would increase. Even without these additional measurements, the location of the measurement site can be used for data interpretation. As a result, studies that consider sea spray as a particle source should consider the effect of humidity on the particle detection efficiency. Furthermore, the enhanced

activation of soluble particles can also be used to improve the detection efficiency for hygroscopic seeds.

These changes have been included in the manuscript:
"Thus, number concentration measurements should be paired with chemical and/or hygroscopicity measurements to improve the assessment of the uncertainty. If these additional measurements cannot be conducted, the location of the measurement site should be considered for data interpretation. Especially when conducting studies in marine surroundings, at which sea spray is one of the contributing particle sources, special care should be taken during the data evaluation (Lawler et al., 2014; Zieger et al., 2017)."

***A concerning factor as to whether these results apply to the atmosphere: Are there actually any nanometer-sized NaCl aerosol in the atmosphere?***

**Response:**
It was selected for our studies, since it is known to be very hygroscopic and well characterized by different studies like Biskos et al. (2006) and Krämer et al. (2000). Clarke et al. (2003) were able to characterize the size distribution of particles in the atmosphere produced by breaking waves. They could demonstrate that those particles cover the size range from 10 nm to greater than 10 µm. Regarding the chemical composition of nanoparticles in marine surroundings, M. J. Lawler et al. (2014) conducted TDCIMS measurements in Mace Head, Ireland. The composition measurement of particle between 15-85 nm showed measurable levels of sodium and chloride. Sea spray is a complex mixture of inorganic salts and organic material and a recent study by Zieger et al. (2017) showed that the hygroscopicity of the inorganic component is about $8 - 15\%$ lower than of pure sodium chloride.

***The discussion of the rearrangement / change in shape of these particles is missing any experimental description. This experimental data is a (another?) humidity effect and seems to be pertinent to the topic here.***

**Response:**
At relative humidity below the deliquescence threshold NaCl particles with an initial mobility diameter between 19-200 nm undergo a microstructural rearrangement as shown by Krämer et al. (2000). Subsequently, G. Biskos et al. (2006) investigated the deliquescence and efflorescence of NaCl seeds with a mobility diameter between 6-60 nm under different humidity conditions using a tandem DMA setup. Both studies revealed a shrinkage with relative humidity conditions below the deliquescence threshold for dry sodium chloride particles. The cut-off diameter of the used TSI 3776 UCPC is at 2.5 nm according to the manufacturer at standard temperature settings. Therefore, we were aiming at studying the size range around the UCPC cut-off diameter using a tandem DMA setup to investigate the shrinkage of NaCl particles when exposed to a defined RH (the setup is also shown in Figure 1 in the comments of the second reviewer). Due to the low particle concentrations below 4.5 nm our measurements were limited to this particle size, but still the microstructural rearrangement yielded an attenuated decrease in size with decreasing particle diameter. Therefore, we assumed that the rearrangement or change in shape plays an important role for seed sizes >3 nm, but below the measured charge enhanced nucleation for sodium chloride particles influence the nucleation and growth process (C. Tauber et al., 2018). As a result, water molecules dissociate the NaCl cluster and a charge enhanced nucleation / growth leads to an enhanced activation.

***The discussion around this shape effect would be greatly helped if data could be presented with the DMA run at the RH of interest. Why was not a recirculating pump / filter not used for some measurements? That would help the interpretation of the overall results and may lead to some sort of basis for a model description.***

**Response:** These changes have been included in the manuscript:
"In order to test a humidity induced change in particle size, we used a standard DMPS setup, which has a filtered and dried (using Silica gel) sheath air loop. The corresponding setup schema can be found in the supporting information (Figure S3). Our results confirmed the measurements conducted by Biskos et al. (2006), showing the shrinkage of sodium chloride particles in the presence of water vapor."
Nevertheless, in the future we would like to conduct these measurements with humidified sheath air for comparison. Additionally, higher RH conditions could be used to investigate the influence of deliquescence on even smaller particles.

*There is previous work on humidity effects for nanometer-sized sulfuric acid particles out of the Eisele-McMurry collaboration at NCAR. Also the O'Dowd group explored chemical effects and the activation of nano-particles. Those two studies were focused on pulse height analysis but the humidity effect explored here is integral to those experiments. I think the Donaldson group out of Toronto also discussed humidity effects in butanol CPCs. There are probably others and they should be cited. How would the PSM (Seinfeld, Kulmala etc.) instrument data be affected? How about alternate condensing fluids like diethylene glycol or FC43 (the latter used on aircraft campaigns by Brock and co-workers) ?*

**Response:**
We acknowledge the well-justified comment and thank the reviewer for making us aware of the pulse height analysis studies.

[revised manuscript text omitted]

*The authors should consider changing the language from inverse temperature to some-thing like non-congruent (with the Kelvin ....) temperature dependence. Inverse temperature to many physical scientist means a 1/T dependence.…*

**Response:**
We thank the reviewer for the comment and reformulated to opposite temperature trend.

*The non-native English speaking authors have had difficulty translating their thoughts into English. Hard to follow their logic at times. More on this at a later time: wanted to get these major comments to the authors for discussion etc.*

**Response:**
The manuscript has been revised concerning grammar and word order.

**Reviewer: 2**

General comments:

*Summary: The paper demonstrates a humidity dependence on NaCl activation using a butanol CPC, where increasing humidity decreases the activation cut-size, while showing no such dependence with Ag particles. The measurement was performed with both continuous flow CPCs and the SANC. Due to the strong humidity-dependence of some particles, it is believed that ambient measurements could be activated at smaller sizes than their laboratory-controlled equivalent and care should be taken in assuming constant cut-sizes for butanol-based CPCs in ambient measurements of unknown composition.*

*These results seem significant and interesting, but I think it could be reorganized to better integrate, introduce, and motivate each experiment's contribution to the objective. Some experiments are not fully motivated of their potential significance until the results section.*

*The abstract/intro is missing a significant section of the paper regarding charge effect. Poor humidity sensor selection for a paper on targeting humidity effects. Sensor uncertainties need to propagate through to resulting figures (e.g. Figure 4).*

**Response:**
A paragraph in the introduction for the charge effect was added in the updated manuscript. The shrinkage measurements in Figure 4 were conducted with SHT75 sensors, with an accuracy of +/- 1.8% and with a HMI38 Humidity Data Processor with a HMP35E probe,with an accuracy of +/- 2.0%. For the sake of readability uncertainties were not included in Figure 4 in the final version of the manuscript. The corresponding uncertainties are discussed in the figure caption instead.

*Would a working fluid such as Fluorinert counteract the humidity dependence of NaCl activation, or would the NaCl still uptake water and nucleate more easily with humidity? Practical solutions and guidelines for scientists would be helpful.*

**Response:**
These compounds consist of carbon chains with fluorine atoms instead of hydrogen atoms. These molecules have a low dipole moment. Due to the spherical structure and very electronegative atoms on the outside the overall surface of the molecule is slightly negatively polarized. As a result, a shift in the cut-off diameter can be expected by using a mixture of Perfluorcarbonates and water. The overall polarity of the working fluid would almost exclusively be based on the dipole moment of water molecules. In fact, the interaction between the NaCl seeds and the water molecules will be persistent if only the working fluid gets changed. To remove the influence the carrier gas should be kept dry. In case the RH cannot be kept completely dry or only dry below a certain threshold value, the effect of a change in cut-off size should be considered in the data analysis, especially at measurement sites located near the sea.

*Major Comments: The citations provided, on nucleation for example (P2 L5), should credit the work from previous authors. Also, heterogeneous nucleation mechanisms needs to include more sources than Wang et al. 2013. In general, the sources need to be expanded to include the major pioneers of the subjects. If the experiments were already performed with propanol in Schobesberger et al., why would a different behavior with butanol be expected? The difference between the papers, aside from changing the working fluid, should be highlighted.*

**Response:**

We acknowledge the well-justified comment on the citations of heterogeneous nucleation mechanism and included them in the updated manuscript. We were not expecting a different behavior. Initially the temperature dependence found by Schobesberger et al. (2010) was used to explain the enhanced detection efficiencies for NaCl found during the laboratory measurements. During the measurements it was found that the relative humidity dependence was stronger than the temperature dependence. Schobesberger et al. (2010) focused on the temperature dependence and compared it to different theoretical approaches like Fletcher theory. In this work we focused on the deviating activation behavior of butanol-based CPCs. The deviating activation behavior is caused by the temperature settings and by increased RH levels.

*Figure 1 does not include control system mechanism for RH. The paper focuses on humidity as a primary variable, but not much detail was provided on how it was varied and controlled. Is RH controlled with a feedback loop to account for transients through-out the experiment? If RH is the main variable of interest, its introduction to the system and control should not be glossed over.*

**Response:**
Figure 1 was moved into the supplemental materials. In the revised manuscript it was replaced by the Figure 1 shown below.

These changes have been included in the manuscript:
"Accordingly, the carrier gas was humidified by passing it through a diffusion type humidifier (see Figure 1). The relative humidity of the carrier gas was monitored and recorded throughout every measurement. The deviation of the recorded values was always below ± 1% which is below the sensor accuracy."

[Figure]

*Figure 1. The experimental setup for evaluating the relative humidity dependent counting efficiency of continuous flow type CPCs (TSI 3776 UCPC), which was measured relative to a Faraday Cup Electrometer (FCE).*

*Honeywell sensors mentioned are crude (accuracy +/- 3.5%, hysteresis 3%, repeatability +/- 0.5%, etc.). These uncertainties need to be reflected in the figures.*

**Response:**

The humidity measurements were conducted with two different humidity sensors. The cut-off diameter measurements were conducted with the Honeywell sensors. Due to the accuracy of +/- 3.5% the minimal RH difference between two measurements was 10%. The shrinkage measurements were conducted with SHT75 sensors with an accuracy of +/- 1.8%, hysteresis 1%, repeatability +/- 0.1% and with a HMP35E probe with an accuracy of +/- 2.0%. In the final version of the manuscript the error bars were not included in Figure 4 for improved readability. Uncertainties will be discussed in the figure caption.

***Regarding the counting efficiency experiment (P4), each TSI CPC can have a unique counting efficiency based on laser age, optics cleanliness, alignment, etc. Characterizing the effect of CPC deltaT on nucleation effectiveness using 3 different CPCs does not seem substantial enough, as each CPC could have a different counting efficiency at the same temperature. It would have been preferable to compare one CPC and repeat the experiment 3 times, so the additional factors mentioned above are constrained. Otherwise, it would need to be stated that the 3 CPCs have been normalized to one another for each of the deltaTs (or proven operation is identical).***

**Response:** These changes have been included in the manuscript:
"In total, about 400 detection efficiency curves were recorded. To rule out any dependence of the individual activation behavior of different CPCs, measurements involving one single CPC set to all three temperature settings one after another, and measurements involving three CPCs that were all set to one specific setting, have been performed. Based on these measurements, the maximum error linked to the individual activation behavior was calculated to be at the highest at 8.1%."

Figure 2 and 3 are chosen as exemplary measurements for different temperature settings and CPCs at the same T settings (C. Tauber et al., 2019) and can be found in the supplemental material.

[Figure]

*Figure 2. NaCl positive with 10% RH, all curves were recorded with the same CPC.*

[Figure]

*Figure 3. NaCl negative with 0% RH, all curves were recorded with different CPCs but at constant temperature settings.*

***The charge effect isn't addressed in the abstract or introduction/background and seems significant to the NaCl activation. The inclusion of this experiment is good, but it feels like it came only at the result of Section 3.1, and didn't have its own proper introduction to motivate why it's included.***

**Response:**
We thank the reviewer for making us aware of the leak in the introduction section. In the updated manuscript we added an additional paragraph on the charge dependence to the introduction.

These changes have been included in the manuscript:
"In a recently published paper by Tauber et al. (2018), it has been shown that the presence of monoatomic ions significantly lowers the energy barrier for the heterogeneous nucleation of n-butanol. It is known from previous studies that ion induced nucleation is highly affected by the chemical structure of the condensing vapor and seed ion properties like charge state (Iida et al., 2009; Kangasluoma et al., 2016). This process takes place in the sub-3 nm size range at which condensation-based nanoparticle detection methods have their lower detection limit. Thereby the initial charge state of aerosol particles with diameters within this size range can influence the detection efficiency."

***Minor Comments: P1 L14-22: This paragraph seems irrelevant to the focus of the paper goals and losing it would not detract from the message. The overview seems a little vague, when it would benefit with a background more targeted to the objective.***

**Response:**
We removed this paragraph and added an introduction paragraph on the charge dependence / effect.

*Commas must be added in sentences using passive voice. This is done with some, but not all sentences and must be corrected throughout.*

**Response:**
We reviewed the manuscript to correct for that.

*Simplify wordy sentences throughout:  e.g.  P2 L4:  "This process is called nucleation and arises..." could simplify to "This process of nucleation arises..."; and P2 L18.*

**Response:**
This change has been included in the updated manuscript.

*P2 L7:  Remove "a" to make CCN plural to agree with your verb and that you used "nuclei" instead of "nucleus", i.e.  "...nanometer-sized particles contribute as cloud condensation nuclei (CCN)."*

**Response:**
This change has been included in the updated manuscript.

*P2 L15-17:  You describe deliquescence and efflorescence without using the term for the mechanism.*

**Response:**
This has been included in the updated manuscript.

*P2 L 20: Subject-verb agreement; "One technique...is CPCs", not "are".*

**Response:**
This change has been included in the updated manuscript.

[revised manuscript text omitted]

---

## Author Response (AR2)

**Authors Response to the Reviewer Comments regarding the manuscript "Humidity effects on the detection of soluble and insoluble nanoparticles in butanol operated condensation particle counters" by Tauber et al.**

Dear editor,

we appreciate the thoughtful comments by the third reviewer which have substantially helped to improve and clarify our manuscript. All comments have been considered and are explained in the detailed point-by-point answers.

Sincerely,

Christian Tauber on behalf of all co-authors

**Reviewer:**

General comments:

*General comments:The manuscript by Tauber and coworkers present an extensive set of experiments probing the heterogeneous nucleation of butanol with NaCl and silver seeds, and influence of water vapor on the process. The used nucleation reactors are the SANC and a commercial TSI 3776 CPCs. Wide range of conclusions are made on the effect of particle charge, composition and hygroscopicity on the nucleation probability. Interesting findings, among many others are that increasing sample flow RH decreases the CPC cutoff for NaCl seeds while not for Ag seeds, which is explained by water vapor uptake by NaCl seeds but not Ag. Contradicting results are presented for the role of charge measured with the SANC (enhancement) and 3776 (suppression) on the nucleation probability of NaCl particles with butanol, which is not satisfactorily explained. I have also other concerns on some interpretations of the data that are specified below. What distracts me also is that the manuscript is full of comparative statements (reduces, increases, lowers etc) without specifying what is compared to what, or otherwise logically incomplete statements so that they could be well understood. Some examples are picked below. It makes the text hard to read and partly ambiguous. Overall, I think the presented data and results are valuable and publishable, but at current state, the manuscript requires improvements before publication.*

**Response:**
We want to thank the reviewer for taking the time to review our manuscript and for his/her positive feedback to our work.

*Below are my detailed concerns.*

*- P2 l12-13, lower the energy barrier compared to what?*
*Response:* This sentence has been rephrased by:
"In a recently published paper by Tauber et al. (2018), it has been shown that the presence of monoatomic ions significantly lowers the energy barrier for the heterogeneous nucleation of n-butanol compared to neutral seeds."

*- P2 l12-17, the particle chemical structure plays major role, as shown for example in each of the three cited references in the paragraph and this study. Also, it is not limited only to sub-3 nm size range as shown for example by this current study (fig6) and many previous water CPC studies*
**Response:** This change has been included in the manuscript
"This process comes into play especially in the sub-3 nm size range at which condensation-based nanoparticle detection methods have their lower detection limit."

*- P2 l24, "indicated a decrease of the activation barrier", because of what?*
**Response:** This sentence has been rephrased:
"Previous studies on binary heterogeneous nucleation of n-propanol-water vapor mixtures on sodium chloride (NaCl) particles indicated a decrease of the activation barrier in accordance with the theory of binary heterogeneous nucleation and reported a soluble-insoluble transition in the particle activation behavior (Petersen et al., 2001)."

*- P2 l25-26, the high relevance is coming from that many CPCs operate with butanol, and not*
**Response:** This change has been included in the manuscript
"Butanol is chemically similar to propanol and commonly used as working fluid in CPCs. Therefore the process of heterogeneous nucleation of n-butanol and water vapor on nanometer sized particles is of high relevance in ambient nanoparticle characterization studies."

*- P2 l28, "revealed a strong dependence on chemical composition and water vapor", what is dependent on chemical composition and water vapor?*
**Response:** This sentence has been rephrased:
"In addition, pulse height analysis conducted by Hanson et al. (2002) for sulfuric acid particles revealed a strong dependence of the particle counter's response on chemical composition and water vapor."

*- P2 l30, affect physicochemical properties of what?*
**Response:** This sentence has been rephrased:
"The size resolved chemical composition measurements of nanoparticles from reactions of sulfuric acid with ammonia and dimethylamine, investigated by Chen et al. (2018), suggest that small, acidic and newly formed particles can affect the physicochemical properties during the cluster formation process and thereby enhance early particle growth."

*- P3 l3-4, in the scope of this study, at least the work of Barmpounis et al. (2018) and Sem (2002) should be relevant for their observations of detection efficiency as a function of condenser temperature and effect of RH, respectively. Possibly also relevant references are Iida et al. (2009) and Kangasluoma et al. (2013) who have studied RH dependence of DEG based CPCs*
**Response:** These sentences have been rephrased and added:
"Despite the common use of butanol as working fluid for the detection of ambient nanoparticles, comparatively little research on the fundamental aspects of butanol nucleation has been done in the sub-10 nm size range, e.g. Barmpounis et al. (2018) and Sem (2002). For DEG based CPCs Iida et al. (2009) and Kangasluoma et al. (2013) investigated the temperature and humidity dependence of the working fluid on the particle activation. Here, we investigate heterogeneous nucleation of n-butanol vapor on differently sized seeds in the sub-10 nm diameter range depending on relative humidity, charge state and nucleation temperature."

*- P3 l6 "we focus on the deviating activation behavior of butanol-based CPCs in this work." What does this mean?*
**Response:** We removed this sentence:
"In contrast to Schobesberger et al. (2010), we focus on the deviating activation behavior of butanol-based CPCs in this work."

*- P3 l20, what kind of ion trap was used?*

**Response:** These sentences have been rephrased and added:

"Subsequently, ions that form inside the charger were removed using a home-built ion precipitator (ion trap). By applying a voltage of +/- 500V to a tubing which is cut in two half pipes and separated by a non conductive material all charger ions were removed."

*- P3, how it was taken care that particle losses in the lines due to diffusion do not affect the results? Are the particle losses the same for the measurements of charged and neutral experiments?*

**Response:** This sentence has been added:

"Diffusional particle losses were kept to a minimum by keeping tubing as short as possible. In every measurement sequence, particle tubing to CPC and FCE was kept to the same length. In addition, the flow rates were kept constant and the flow was symmetrically split. The additional particle losses for the neutral experiments were calculated following Tauber et al. (2019)."

*- P4 l10-19, if you neutralize the size selected particles, what is the concentration reference for the neutral particles? It cannot be the FCE as in eq1.*

**Response:** This sentence has been added:

"In order to calculate the counting efficiency of the neutralized aerosol, we evaluated the charging efficiency and the theoretical penetration efficiency for the FCE and CPC to correct for the neutralization and penetration losses following Tauber et al. (2019)."

*- P5 l21-22, can you speculate why is this? If silver is generally considered insoluble, why it is even harder to activate sodium chloride than silver? Is it related to the NaCl particle shrinkage due to water (see next comment)? Also, can you elaborate somewhere briefly in the manuscript why Kelvin predicts increasing onset S for decreasing Tnuc? To me it is not obvious*

**Response:** According to our findings the particle rearrangement due to the increased water vapor under high RH conditions supports the nucleation process for hygroscopic NaCl seeds but not for hydrophobic Ag particles. As a result, with increasing RH NaCl seeds can be activated easier than Ag seeds. The two systems have quite different situations: Ag particles are insoluble and hydrophobic, so they do not care much about sub saturated water vapor (see Petersen et al., 2001). It's rather unary butanol nucleation in the presence of water vapor. NaCl particles on the other hand are soluble in water but insoluble in butanol. For pure butanol nucleation (low RH) on NaCl the difference to Ag particles can be attributed to structural differences of the seed surfaces (crystalline vs. spherical) and different seed-liquid interactions.

This sentence has been rephrased:

"This is in line with classic Kelvin predictions (Thomson, 1871), where with decreasing temperature the required equilibrium saturation ratio increases. This behavior agrees with homogenous nucleation."

*- P9 l10-16, does this speculation mean that the NaCl-butanol measurements with RH < 3.5% and NaCl-propanol measurements of Schobesberger et al. (2010) are also contaminated with water? And that for < 3.5 nm sizes the seed is just too small to uptake water at lower RH, and the unusual trend vanishes or is not observed at all (2.5 nm)? Is this also why at RH 0% in fig4 the normalized mean diameter is never 1? Due to the previous, to me it looks that the RH < 3.5% data is also affected by water vapor, and the discussion should be modified accordingly.*

**Response:** Schobesberger et al. (2010), also conducted an assessment for possible water contamination but with the result that the water enrichment in the dry was always below the sensor accuracy at low RH conditions. We can only speculate that in this study water vapor could have been an effect on the measurement results. In order to reduce water contamination we used bottled air in contrast to Schobesberger et al. (2010). So our data were most likely taken at lower water

vapor concentrations. For larger particle diameters the reversed trend did not vanish and confirms Schobesberger et al. (2010).

Due to experimental limitations we could not reach dry RH values of 0%, this is also one reason why the normalized mean diameter is never 1. We assume, particles with a mobility diameter < 3.5 nm are less affected by the water vapor. In this size range the charge state and chemical composition maybe more important.

*- P11 l1, there is structural change according to fig4 even at RH 3.5%*

**Response:** This sentence has been rephrased:

"For comparison, particle classification with the nDMA was always performed under dry conditions (RH < 3.5%), so that the structural change of the sodium chloride clusters resulting from water vapor is kept to a minimum."

*- P11 l8, is the number 8.1% for a single detection efficiency value or cutoff or something else?*

**Response:** This value is the maximum deviation of the measured cutoff.

*- P11 l12, this seems to be true around RH > 20% according to figs 6-7, while not true according to fig2 where you have RH < 10%*

**Response:** In Figure 2, we show the onset saturation ratio under different nucleation temperatures for different mobility sizes. The results show a decrease of the onset saturation ratio with decreasing nucleation temperature for sodium chloride seeds and vice versa for Ag particles. To investigate the nucleation temperature dependence of NaCl seeds we measured the same mobility diameters under low RH conditions (RH < 3.5%) and compared it with the measurements conducted under elevated conditions (RH < 10%) as shown in Figure 3. As a result, the onset saturation increases with decreasing RH for the larger seeds. Consequently, the NaCl particles can be easier activated than an insoluble seed.

*- P11 l15, "decrease", compared to what? To RH 3.5% data? Based on fig3 I would say increase*

**Response:** This sentence has been rephrased:

"These measurements were conducted under elevated humidity conditions (RH = 10% at room temperature), which lead to a decrease compared to dry conditions (RH=3.5%) of the necessary onset saturation ratio for neutral NaCl seeds as shown in Figure 3."

*- P11 l22, "charge history", with respect to this study I think this term is misleading. In an ideal experiment the generated positively and negatively charged particles are identical except for the charge sign. When they are neutralized, ideally the resulting neutral particle is the same, and thus also its activation related properties, regardless of the initial charging state. The reason why different results are obtained with different "charge histories" can be either that the particles are different from the beginning, or that their properties change during the neutralization process. Kangasluoma and coworkers have shown some evidence for the former, that the composition of the generated particles are different in different polarities (and also that it affects the cutoffs of some CPCs) e.g. in the paper referred below, and in the paper cited in the manuscript. For the latter Alonso and Alguacil (2008) have provided measurements. Therefore, I suggest removing the term "charge history", and add some other speculation to explain the observations, or at minimum elaborate clearly the mechanism how "charge history" can affect the experimental results.*

**Response:** This sentence has been added:

"An effect from charge history could possibly be that previously positively charged particles differ from those that were negatively charged prior to neutralization, e.g. by their original properties or because of a change of properties during the neutralization process (Kangasluoma et al., 2013; Alonso and Alguacil, 2008)."

*- P11 l27-31, to me this speculation is contradictory to previous. First it is speculated that solvation of NaCl to water decreases the onset S of butanol, while here it is speculated that increased water uptake of charged seeds increases onset S of butanol (cutoff)*

**Response:** Here we do not conclude that the onset saturation ratio is increasing with higher water vapor. But the charged NaCl seeds may attract more water molecules under sub-saturated conditions than a neutral seed. Thereby the particle dissociation is faster for charged particles which would lead to smaller sizes and lower detection efficiencies/larger cut-off sizes.

*- P11 l35, polarity of the cluster, or polarity of the vapor molecule?*

**Response:** This sentence has been rephrased:

"Increasing amounts of water molecules are mixed with already preexisting butanol molecules. As a result the overall bulk polarity of the working fluid increases and the activation of sodium chloride clusters is improved."

*- P12 l2-9, These statements need more elaboration. With SANC charge enhancement is observed at <= 3.5 nm, while with CPCs neutrals are activated more easily more or less at all measured sizes (1.5-4.5 nm). So I think the data does not support that the particle size can explain the observed differences between the SANC and CPC. Also, I don't understand how the temperature trend affects the charge enhancements? The processes should be the same for both instruments? L5-9 are hard to understand*

**Response:** These sentences have been rephrased:

"This is probably due to the cut-off diameter size which is for most of the counters > 3 nm. Except for negatively charged NaCl particles, which show a charge enhancement as it was observed with the SANC for particles < 3 nm mobility diameter. In addition, the recorded cut-off diameter for high T settings is in the size range, where the opposite temperature trend was recorded by the SANC measurements. Consequently, by decreasing the temperature the needed saturation ratio to activate a NaCl particle in this size range is decreasing. Furthermore, the increasing RH supports the structural rearrangement which reduces the cut-off diameter (see Figure 6) compared to dry conditions. As stated in Castarede et al. (2018), the solvation of sodium chloride clusters leads to an electrical potential. In this study, we show that nucleation of n-butanol vapor onto soluble NaCl below 3.5 nm is enhanced. This effect takes place as the aerosol is exposed to RH, i.e. before the sample enters the CPC. Subsequently, inside the CPC, the RH increases due to the temperature reduction inside the condenser which will have an effect on the charge enhancement. As a matter of fact, our results lead to an improved detection efficiency for sodium chloride clusters. At high relative humidity and low temperature settings the detection efficiency is further promoted by charge enhanced nucleation."

*- P13 l6-8, why is this? It is exactly the opposite what is reported by Winkler et al. (2008). Also, why saturation vapor pressure would affect charge enhancements, instead of e.g. supersaturation?*

**Response:** These sentences have been rephrased:

"Apparently, the particle charge only supports the nucleation and growth process under high n-butanol and water (black) saturation vapor pressure conditions i.e. low supersaturation conditions (high CPC Temp settings, see Table 1). In fact, this finding supports the study of Winkler et al. (2008). They proposed that the needed onset saturation ratio for charged particles is lower than for neutral particles. Under high CPC temperature conditions (high T settings), saturation ratio is the lowest. The nucleation barrier which has to be overcome to activate charged seeds at the size of the cut-off diameter is therefore lowered by charge enhancement. As a result, the charged silver particles can be activated but not the neutral Ag seeds under low saturation ratio conditions (black). According to the SANC measurements, no charge enhanced nucleation for positively charged Ag particles could be observed. Similarly, we do not find substantial cut-off diameter reduction for positively charged Ag particles. Only at the lowest temperature settings a sign-preference becomes visible favoring the activation of negatively charged particles."

*- P14 l1-3, what do you mean? Why solubility has something to do with high concentration?*
*Response:* We removed this sentence:
"Due to high particle number concentrations in the nucleation mode, atmospheric measurements of soluble and insoluble nanoparticles might lead to wrong data interpretation."

*- P14 l8-9, can you recommend a good RH threshold? If I remember right ACTRIS suggests RH < 40%, is it good?*
*Response:* We have no recommendation for inlet flow humidity. Except that, in the ideal case, measuring instruments are calibrated up to the maximum RH that they are exposed to. The suggested RH threshold of <40% is reasonable, but the recommendation should also take into operating temperatures of the used CPCs.
This sentence has been added:
"In the ideal case, measuring instruments are calibrated up to the maximum RH that they are exposed to.*"*

*- P14-15 l10-5, are you suggesting sub-10 nm particles in the coastlines are NaCl from sea spray? Can you provide some references? I think so far iodine and sulfuric acid have been shown to be responsible for sub-10 nm particles in the coastlines (Jokinen et al. 2018; Sipilä et al. 2016). What are hygroscopicities of these particles? Also, how would you suggest to correct for this? Adjust the CPC cutoff, or also somehow correct the measured concentration?*
*Response:* These sentences have been rephrased and added:
"Furthermore, studies that investigate aerosol from sources that involve soluble seeds, should consider the effect of humidity on the particle detection efficiency. Soluble seeds do not only include sodium chloride particles which were analyzed in this study. Ammonium sulfate, which originates from the gas-phase from precursor gases having both natural and anthropogenic origin (Sakurai et al., 2005; Smith et al., 2004), can act as soluble seed. Consequently, we suggest to consider a possible shift in cut-off diameter when such soluble species are present in the sample."

*- P15 l9-10, is it (Kangasluoma et al. 2013)?*
*Response:* This sentence has been added:
"As presented in Kangasluoma et al. (2013) variation of the inlet flow RH leads to increased detection efficiency for a turbulent mixing type CPC."

*- P15 l10-12, DEG has been shown to be good in activating 1 nm particles in a CPC modified from 3776 (there are many papers on DEG based ultrafine CPC modifications), so it is not the "highly dynamic environment" that makes PSMs good down to 1 nm.*
*Response:* This sentence has been removed:
"As a result, it can be inferred that such a shift in the cut-off diameter is expected to be smaller than in the case of butanol. The working principle of PSMs (turbulent mixing) establishes a highly dynamic environment that leads to an increased rate of particle activation even without the presence of water."

*- P15 l18, which size change?*
*- P15 l18-19, I don't understand this sentence*
*- P15 l19, which influence?*
*Response:* These sentences have been rephrased and added:
"Pulse height analysis measurements by Hanson et al. (2002), revealed a pronounced substance dependency and a difference in peak distribution when using n-butanol or FC-43 as working fluid in a CPC. The interaction between NaCl seed and water molecules remains regardless of the working fluid. If RH cannot be kept at 0%, the instrument should be calibrated to the corresponding

conditions and a possible change in cut-off diameter be considered in the data analysis to ensure high data quality."

*- P15 l29-30, this is not true according to figs 6 and 7*
*Response:* These sentences have been rephrased and added:
"However, for negatively charged NaCl seeds a charge enhanced activation was measured with the SANC. Above 3.5 nm the charge does not play a role during the nucleation process. Charge enhanced particle activation did only occur at the lowest CPC temperature settings."

*- P16 l1-2, which polarity? There are several other similar studies on the topic. It is hard to follow the logic of this and the following sentence*
*Response:* These sentences have been rephrased and added:
"A recently published study on heterogeneous nucleation of n-butanol on monoatomic ions has shown that the initial formation of a critical cluster does not favor one of the two possible polarities of the seed particle. However, under sub-saturated conditions the existence of seed particle charge plays and important role concerning vapor uptake."

*- P16 l14-16, this is contradictory to fig6 where neutrals are activated more easily than charged*
*Response:* These sentences have been rephrased and added:
"If water vapor is present charge effects on the surface promote the initial steps of nucleation even more. As a result, the energy barrier for activating nanometer sized NaCl particles can be reduced - this is comparable to already published results for n-propanol/water mixtures conducted by Petersen et al. (2001)."

[revised manuscript text omitted]

---

## Author Response (AR3)

***Authors Response to the Reviewer Comments regarding the manuscript "Humidity effects on the detection of soluble and insoluble nanoparticles in butanol operated condensation particle counters" by Tauber et al.***

Dear editor,
we appreciate the help and thoughtful comment by the third reviewer to improve and clarify our manuscript. The last comment have been considered and is explained below.

Sincerely,
Christian Tauber on behalf of all co-authors

*Reviewer:*

*Comment:*
*- P13 l6-8, why is this? It is exactly the opposite what is reported by Winkler et al. (2008). Also, why saturation vapor pressure would affect charge enhancements, instead of e.g. supersaturation?*

*Response: These sentences have been rephrased:*
*"Apparently, the particle charge only supports the nucleation and growth process under high nbutanol and water (black) saturation vapor pressure conditions i.e. low supersaturation conditions (high CPC Temp settings, see Table 1). In fact, this finding supports the study of Winkler et al. (2008). They proposed that the needed onset saturation ratio for charged particles is lower than for neutral particles. Under high CPC temperature conditions (high T settings), saturation ratio is the lowest. The nucleation barrier which has to be overcome to activate charged seeds at the size of the cut-off diameter is therefore lowered by charge enhancement. As a result, the charged silver particles can be activated but not the neutral Ag seeds under low saturation ratio conditions (black). According to the SANC measurements, no charge enhanced nucleation for positively charged Ag particles could be observed. Similarly, we do not find substantial cut-off diameter reduction for positively charged Ag particles. Only at the lowest temperature settings a sign-preference becomes visible favoring the activation of negatively charged particles."*

*Follow up comment:*
*Fig7 shows that charge lowers the cutoff clearly at low supersaturation settings of the CPC (the biggest difference in cutoff between charged and neutral), but not at high supersaturation settings. Winkler et al 2008, on the other hand, shows the biggest difference in heterogeneous nucleation probability (that is in this respect quite the same as cutoff) between the charged and neutral at high supersaturation, while minimal charge effect at low supersaturation. I still find these two observation contradicting.*

*Response:*
We want to thank the reviewer for taking the time to review our manuscript and for his/her positive feedback to our work.

*- P13 l27- P14 l2:*
*Response:* This sentence has been rephrased by:

[revised manuscript text omitted]